# Addressing Immune Response Dysfunction in an Integrated Approach for Testing and Assessment for Non-Genotoxic Carcinogens in Humans: A Targeted Analysis

**DOI:** 10.3390/ijms26136310

**Published:** 2025-06-30

**Authors:** Annamaria Colacci, Emanuela Corsini, Miriam Naomi Jacobs

**Affiliations:** 1Agency for Prevention, Environment and Energy, Emilia-Romagna (Arpae), Via Po 5, 40139 Bologna, Italy; 2Alma Mater Institute on Healthy Planet, University of Bologna, Via Massarenti 11, 40138 Bologna, Italy; 3Department of Pharmacological and Biomolecular Sciences ‘Rodolfo Paoletti’, Università degli Studi di Milano, Via Balzaretti 9, 20133 Milano, Italy; emanuela.corsini@unimi.it; 4UK Health Security Agency, Radiation, Chemical, Climate and Environmental Hazards, Harwell Innovation Campus, Chilton, Oxon OX11 0RQ, UK; miriam.jacobs@ukhsa.gov.uk

**Keywords:** AhR, cytokines, immune evasion, immune suppression, immunotoxicity, IATA, test methods, new approach methodologies

## Abstract

Most known chemical carcinogens induce the direct activation of DNA damage, either directly or following metabolic activation. However, carcinogens do not always operate directly through genotoxic mechanisms but can do so via non-genotoxic carcinogenic (NGTxC) mechanisms. Immune dysfunction is one of these key events that NGTxCs have been shown to modify. The immune system is a first line of defence against transformed cells, with an innate immune response against cancer cells and mechanisms of immune evasion. Here, we review the key events of immune dysfunction. These include immunotoxicity, immune evasion, immune suppression and inflammatory-mediated immune responses, and the key players in the molecular disruption of immune anti-cancer molecular signalling pathways, particularly those mediated by cytokines and the Aryl hydrocarbon Receptor, in relation to the identification of NGTxC. The plasticity of cytokines towards functional flexibility in response to environmental stressors is also discussed from an evolutionary heritage perspective. This is combined with a critical assessment of the suitability for the regulatory application of currently available test method tools and is corroborated by the key biomarkers of, e.g., MAPK, mTOR, PD-L1, TIL and Tregs, CD8+, FoxP3+, WNT, IL-17, IL-11, IL-10, and TNFα, as identified from robust cancer biopsy studies. Finally, an understanding of how to address these endpoints for chemical hazard regulatory purposes, within an integrated approach to testing and assessment for NGTxC, is proposed.

## 1. Introduction

Cancer is a multi-step process which involves initiation, promotion, and progression associated with the alteration of multiple pathways [1,2,3,4]. Cancer can arise from a combination of innate factors, such as genetic susceptibility and chronic inflammation, as well as external factors, including environmental exposure to carcinogens and infectious agents [5]. These factors contribute to the complex interplay of genetic, epigenetic, and environmental factors that underlie the multifaceted process of carcinogenesis.

The role of environmental immune disruptors and inflammation in cancer risk underscores the intricate relationship between environmental exposures and immune responses in cancer development [6]. Chemicals, including environmental factors, are believed to be responsible for the significant increase in diseases over the past few decades, such as cancer (i.e., breast, leukemia, lung, and prostate cancer), allergies, and autoimmunity (i.e., rheumatoid arthritis), which can all be linked to immune system deregulation. Operating in genetically predisposed individuals, these factors may directly initiate, facilitate, or exacerbate the pathological immune processes.

Host immunity is an important barrier to tumor formation and progression [7], and immunosuppression is recognised as one of the key characteristics/hallmarks commonly exhibited by human carcinogens [8,9,10,11]. The immune system plays a key role in cancer initiation, development, and progression. Multiple pathways are involved in evading innate and adaptive immune responses, with a broad spectrum of chemicals having the potential to adversely influence immunosurveillance [7,12].

Most chemical carcinogens induce direct DNA damage, either directly or following metabolic activation. However, carcinogens do not always operate directly through genotoxic mechanisms but can do so via non-genotoxic (NGTx) mechanisms. Non-genotoxic carcinogens (NGTxCs) are defined as chemicals that have the potential to induce cancer without interacting directly with either DNA or the cellular apparatus involved in the preservation of the integrity of the genome [13]. Among the processes that NGTxCs have been shown to modify, endocrine modification, immunosuppression, tissue-specific toxicity, and unresolved chronic inflammation are well recognised [14]. In an analysis conducted by Hernandez et al. [14], approximately 12% (45 out of 371) of IARC Groups 1, 2A, and 2 B carcinogens were acting via NGTx mechanisms.

Traditionally, NGTxCs have been detected in the 2-year rodent carcinogenicity bioassays (RCBs) [15,16]. The test strategy for carcinogenicity starts with the assessment of genotoxic endpoints under in vitro and in vivo conditions. Genotoxicity testing, if consistently positive, means that the chemical in question is considered a genotoxic carcinogen. However, a borderline classification or consistent negative result means that whilst the chemical is not a genotoxic carcinogen, it may induce cancer via non-genotoxic carcinogen mechanisms. The need to address NGTxCs is well recognised and has become the basis for the consensually agreed development of the Organisation for Economic Cooperation and Development (OECD) project, established in 2016, developing an Integrated Approach for Testing and Assessment (IATA) for NGTxCs [13].

The IATA, and the modular approach for the assessment of NGTxC, is a grouping format to organise the mechanisms, Molecular Initiating Events (MIEs) and Key Event (KE) components leading to non-genotoxic carcinogenicity by addressing early (molecular) to mid (cellular) to later (tissue) biological KEs [17]. These are aligned according to increasing adversity and specificity associated with cancer initiation, promotion/progression, and tumor formation. Following a negative outcome from the genotoxicity and mutagenicity regulatory test methods, the workflow assesses existing information, including the use of relevant in silico tools. The first module assesses molecular and cellular mechanisms. The next and central module consists of the four pivotal and parallel KEs of inflammation, immune response, mitogenic sequencing, and cell injury. These KEs are not sequential to each other and can individually, or in combination, lead to the two principal next KEs, (sustained) proliferation [18] and a change in morphology [19].

Here, we review the KE of immune dysfunction (including immunotoxicity, immune evasion, and immune suppression) and inflammatory-mediated immune responses in relation to the identification of NGTxC, together with an assessment of the test method tools that can address these endpoints. This is underpinned by an understanding of the evolutionary routes of immune system development in response to environmental stressors. Parallel efforts with respect to the identification of key characteristics of immunotoxicity are considered to be: Alterations of cell signalling, immune cell proliferation, immune cell differentiation modulation, communication changes between immune cells, and interference with immune cell trafficking [12]. These all fall under the key hallmarks identified for NGTxCs [7,13].

## 2. The Role of the Innate and Adaptive Immune Responses in Cancer Immunosurveillance

The immune system is characterised by two main types of responses: (1) the innate immune response, which is the first line of defence and is considered a non-specific or immediate response, and (2) the adaptive immune response, which is a highly specific and targeted response that develops over time [20]. The innate response activates cytokines such as IL-1, IL-6, IL-8, and TNF-α, whereas the adaptive immune response involves the activation and coordination of different lymphocyte subsets, including T lymphocytes and B lymphocytes and several cytokines that are involved in the differentiation of T cells [21].

The activation and differentiation of T cells are mediated by polarising cytokines, which originate from several types of specialised T cells, including Th1, Th2, Th17, and Treg, each of which produces specific cytokines playing different roles in immune defence [21].

Cytokines can be classified in several ways, depending on their roles in various biological processes [22]. Here, we have grouped cytokines based on their immune function (e.g., pro-inflammatory vs. anti-inflammatory), origin (such as Th1 vs. Th2 cytokines), and involvement in regulatory pathways (such as primary cytokines vs. second-order cytokines) [23]. The term second-order cytokines usually refers to cytokines that are induced by other cytokines during the immune response, rather than being produced directly by antigen-specific T helper cells such as type 1 or type 2 cytokines. These cytokines are often involved in cytokine cascades or loop amplification. For example, cytokines such as IL-6, IL-10, and IL-12 can be considered second-order cytokines because their production is often triggered by primary cytokines (e.g., IL-1 or TNF-αIt is important to consider with respect to the classification of second-order cytokines when identifying biomarkers of immune regulation and dysfunction in cancer). Their regulation is essential in the tumor microenvironment (TME), where a delicate balance between immune activation and suppression can dictate tumor progression or regression. Thus, focusing on second-order cytokines can provide deeper insights into the immune-modulatory mechanisms at play in cancer, helping to better define potential biomarkers for predicting immune dysfunction or therapy responses in oncology.

The second-order cytokines identified originally included only IFNs; IL-10, one of the most potent anti-inflammatory cytokines; and signal transducers and activators of transcription (STATs) and Janus tyrosine kinases (JAKs). However, IL-10 can be considered both a Type II cytokine (based on its role in Th2 responses) and a second-order cytokine, when looking at its production induced by other immune signals. The second-order cytokines activate STAT3 and STAT1, and some also activate STAT2, which is primarily associated with antiviral responses, and STAT5, which is crucial for lymphocyte proliferation and survival. In addition to the canonical signalling route, certain receptors, such as type I interferon receptor (IFNR) and interleukin-22 receptor (IL-22R), can also engage mitogen-activated protein kinase (MAPK) cascades, including those mediated by the ERK, JNK, and p38 kinases.

Accumulating evidence shows that each Th cell type and related second-order cytokines play different roles at the cellular and tissue levels, eliciting specific responses [21], as shown in Figure 1.

## 3. The Bivalent Role of the Immune System in Cancer Development

The immune system plays a key role in the entire cancer process from onset to progression. Several reports over the last few decades have demonstrated the dual behaviour of the immune system, which can either destroy tumors or promote tumor development.

The process by which the immune system identifies and destroys nascent tumor cells is known as cancer immunosurveillance. For the immune system, a tumor can be considered a foreign entity, and in consideration of this, the first line of defence is an innate response. Indeed, the innate immune response may be involved in the initial detection of tumors via mechanisms such as the recognition of cellular abnormalities or local inflammation. However, an adaptive immune response involving antigen-specific B and T cells is often crucial for the control and elimination of cancer [24,25]. This dynamic is visually summarised in Figure 2, which depicts the complex interactions between immune cells and tumor cells within the TME. In the arms race between cancer and the immune system, cancer cells sabotage tumor-infiltrating lymphocytes (TILs) by transferring defective mitochondria, depleting their energy, impairing their function, and ultimately pushing them towards self-destruction and sustaining cancer infiltration [26].

Immunotherapy approaches have been developed to stimulate or enhance the adaptive immune response against cancer, for example, using monoclonal antibodies or immune checkpoint inhibitors [27].

The process by which the immune system promotes cancer progression is often termed ‘immunosuppression’; however, key aspects might also be considered more accurately as ‘immune evasion’. Cancer progression is only possible when tumor cells override immune surveillance [28]. Indeed, cancer development is the result of a complex interplay between tumor cells, normal stromal cells, and host immune defence [29]. This dual role of the immune response is determined, in part, by the interplay between innate and adaptive immunity [30]. Immunosuppression, immune evasion, and immunotoxicity are fundamental aspects of cancer and play a crucial role in disrupting biological mechanisms and processes that can contribute to NGTxC. They are distinctly different immune perturbation or modulation mechanisms and are explained in Box 1.

Box 1**Immune dysfunction terminology definitions.** 
**Immunotoxicity**: Immunotoxicity refers to the adverse effects of chemicals or substances on the immune system. This can include direct toxicity to immune cells and the disruption of immune function, resulting in increased susceptibility to infections, cancers, or inappropriate immuno-stimulation, leading to hypersensitivity or autoimmune disorders due to chemical exposure [31].**Immunosuppression**: Immunosuppression refers to the suppression or dampening of the immune response. This can occur naturally or be induced by medications or treatments to prevent the rejection of transplanted organs or manage autoimmune diseases. In the context of cancer, immunosuppression can also occur as a result of the tumor microenvironment, where cancer cells and associated immune cells produce factors that inhibit immune responses against the tumor [4].**Immune evasion**: Immune evasion in cancer refers to the ability of cancer cells to avoid detection and destruction by the immune system. Cancer cells can employ various strategies to evade immune surveillance, such as downregulating molecules involved in antigen presentation, expressing immune checkpoint molecules to inhibit T cell activation, or secreting immunosuppressive factors to create an immunosuppressive environment around the tumor [3].**Immuno-oncology**: The study of the mechanisms behind cancer initiation and development with the aim of discovering potential therapies to prevent or stop cancer from evading the immune system [32].


Cancer development is generally blocked by CD4+ helper T1 (Th1) and CD8+ cytotoxic cells through the production of IFN-γ and cytotoxins [33]. Tumor cells that present tumor antigens can be attacked and destroyed by immune cells. This requires not only the generation of cancer-specific T cells but also the contact of these T cells with cancer cells [34] (Figure 2). It is during this process that the cancer cells introduce defective mitochondria. Then, with cancer progression, innate immune cells polarise with immunosuppressive properties and, as a result, orchestrate a tolerogenic niche that interferes with the cytotoxic potential of tumor antigen-specific T cells [3,25,35].

In relation to NGTxCs and the role of immune dysfunction, our focus here is on immunosuppression and immune evasion. The ability of a tumor to escape destructive immunity can be ascribed to either innate or adaptive immune evasion. To eliminate malignant cells, tumor antigen-activated T lymphocytes must first pass through the TME and infiltrate the tumor bed.

In innate immune evasion, T cells are excluded from the TME (Figure 2) [36]. In adaptive immune evasion, proliferating cancer cells can accumulate genetic changes, leading to reduced immunogenicity and the appearance of immune-resistant variants [36]. The selection of tumor variants resistant to tumor effectors, which is referred as immunoediting, is characterised by three phases, known as the ‘three Es of cancer immunoediting’; Elimination’, which is supported by CD8+ and NK cells; ‘Equilibrium’, involving Th1 cells, IL-12 and IFN-g, and finally; ‘Evasion’ [25,37,38].

During the elimination phase, the immune system effectively targets and destroys nascent tumor cells. However, as tumors evolve, they may enter a state of equilibrium, where they coexist with immune cells but are not eliminated. The equilibrium phase represents the immune-mediated tumor dormancy, an apparent balance between tumor growth and immune recognition and control [39]. Tumor cells may permanently remain dormant by the induction of senescence, which is considered to be irreversible [39]. Eventually, tumors can escape immune surveillance entirely, leading to disease progression and metastasis. In response to immune pressure, tumor cells evolve in unison to avoid detection by the immune system [40,41]. Several mechanisms can cooperate to sustain tumor cell escape, including reduced immune recognition, due to the loss of tumor antigens; increased resistance or survival, due to the increased expression of anti-apoptotic molecules; and the development of an immune-suppressive TME, mediated by regulatory T cells (Tregs), which represent the major mechanisms of tumor immune evasion (Figure 2 and Figure 3) [42,43].

Tregs are a specialised subset of T cells that regulate immune responses and maintain immune homeostasis. In health, various subsets of Treg cells control the immune response. The Tregs created in the thymus are known as natural Tregs, and those formed in the periphery in response to various tolerogenic stimuli are referred to as induced Treg cells or iT(reg).

Natural Tregs are characterised by the expression of the transcription factor Forkhead box protein 3 (Foxp3), which is essential for their development and function. These cells maintain self-tolerance and prevent autoimmune diseases [44]. Induced Tregs are formed in the peripheral tissues from naïve CD4+ T cells under specific conditions, such as the presence of transforming growth factor-beta (TGF-β) and retinoic acid [45]. The plasticity of iTregs allows them to adapt to various immunological contexts, which is essential for their role in immune regulation.

The dual role of the immune system in both tumor suppression and promotion, as summarised in Figure 3, highlights the complexity of immune-mediated responses in cancer development.

## 4. A Historical Perspective on the Contribution of Immunotoxicity in the Identification of Non-Genotoxic Carcinogens

### The Development of the Regulatory Hazard Assessment of Immunotoxicity

The field of immunotoxicology emerged as a distinct discipline in the early 1970s, driven by observations that exposure to certain environmental chemicals and pharmaceutical agents could impair immune function and increase vulnerability to infections and cancers [12]. Immunotoxicity can manifest in a variety of ways, with one of the most prominent effects being immunosuppression. Research in the 1980s to 1990s led to the development of specific assay tools and the identification of sensitive endpoints that measure effects on the immune response [46,47,48], Table 1. This work was utilised by many regulatory bodies to develop dedicated corresponding immunotoxicity assessment within test guidelines [49,50], including incorporation into in vivo OECD Test Guidelines (TGs) such as the 28- and 90-day studies TG 407 and TG 408 ([51,52], respectively) and the extended one-generation in vivo assay, TG 443 [53].

Two pivotal reports published by the National Toxicology Program (NTP) in 1992 [47] and 1993 [48] significantly contributed to guiding the evolution of immunotoxicology testing guidelines for both drugs and environmental chemicals. Drawing on a dataset of 50 substances, a meta-analysis was conducted to refine future testing approaches, inform hazard assessment, and explore the association between immune function assays and host resistance models. Notably, at that time, just three immune endpoints—antibody production of T cell-dependent antigens, NK cell activity, and surface marker expression—were sufficient to identify immunotoxicants in rodents with full concordance (100%) [47]. Furthermore, a strong correlation was observed between altered immune responses and impaired host resistance [48], with no cases of impaired host resistance occurring independently of immune test alterations, although the reverse was occasionally seen.

In the early 2000s, efforts were made to introduce tiered testing to better incorporate in vitro immune-related assessment tools into testing strategies [55], and current practice for building the evidence basis for industrial agrochemical dossiers is essentially built upon these approaches. In 2017, Corsini and Roggen [56] proposed an in vitro testing strategy for immunotoxicity, reflecting that there are still gaps to fill with respect to the in vitro tests that can be used. Considering the complexity of the immune response, the key targets to study are chemical-induced immunosuppression, which should include the evaluation of effects on bone marrow and innate and adaptive immunity.

In the 1990s, on the basis of analyses of 27 chemicals with sufficient data on the relationship between immunotoxicity and carcinogenicity (Table 2), it was suggested that potential immunotoxic compounds are likely to be rodent carcinogens, with an 81% concordance (*p* = 0.019), 79% sensitivity, and 100% specificity.

The interdependence between immune and host resistance was also examined employing Kendall’s correlation [48], and regarding cancer models, for example, increased susceptibility to syngeneic PYB6 fibrosarcoma tumor cell challenge in mice was associated with a decrease in NK cell activity and cell-mediated immune responses (i.e., mixed leukocyte response, cytotoxic T lymphocytes, and ConA induced T cell proliferation), with no correlation with antibody response or B cell proliferation. The mixed leukocyte response evaluates the compatibility between a donor and a recipient, and as a consequence of the incompatibility of the allogenic determinants on the surface of the two lymphocyte populations resulted in lymphoproliferation. This assay is very useful for assessing lymphocyte function, as proliferation is central to the activation of acquired immunity. Similarly, the assessment of cytotoxic T lymphocytes is important for the evaluation of cell-mediated immunity. Cytotoxic T lymphocytes are a population of CD8+ lymphocytes characterised by specific cytotoxicity for target cells in an MHC-restricted manner. Cytotoxic T lymphocytes are able to destroy a variety of targets, including tumor cells and virally infected cells, which are important for the in vivo elimination of neoplastic transformed cells. T cell proliferation stimulated by mitogens (e.g., concanavalin A, ConA) is an in vitro correlate of the activation and proliferation of specifically sensitised lymphocytes by antigens in vivo. These in ex vivo/in vitro assays have been used clinically to assess cellular immunity in patients suffering from immunodeficiencies, cancer, and autoimmunity, as well as in patients undergoing immunotherapy, and in experimental models to assess the immunotoxic potential of xenobiotics.

Thirty years ago, to evaluate quantitative relationships with respect to the usefulness of the tests available at that time, and to understand how to interpret immune functional changes and carcinogenic process implications, cyclophosphamide was used as a reference immunosuppressive chemical to generate immune–host-resistance relationships. For example, data showed that a 5% relative decrease in PYB6 tumor resistance was associated with a 9.6% decrease in cytotoxic T lymphocytes [48], highlighting an approximate linear model relationship. Table 2 provides a previously published 1992 collation of selected chemicals used in analyses of immunotoxicology relationships [47].

## 5. Towards Integrating Adverse Immune Events into an IATA for Non-Genotoxic Carcinogens

The OECD expert group has identified specific transcriptomic immune markers that can be used for pre-screening, including the aberrant signalling of interleukin and interferon, TNF-α, and JAK-STAT2 [57]. Whilst IL-6 has been identified as a key endpoint for immune dysfunction in relation to NGTxC, to date, only IL-2 and IL-8 luc assays, for different endpoints, skin sensitisation and immunotoxicity, have been developed as OECD TGs so far, in the OECD Test Guideline Programme [58,59]. An OECD detailed review paper on immunotoxicity has been published [60]. Notably, it does not specifically address in vitro test methods for NGTxC-relevant immunosuppression and immune evasion. The gaps identified and acknowledged in the OECD 2022 review are related to immunosuppression, with a need for different immune cell targets and further investigation to distinguish between biomarkers representative of immunosuppression. They are also related to addressing immunosuppression and immune evasion in relation to carcinogenicity and especially NGTxC, which this paper aims to address.

Efforts are underway to develop in vitro assay tools that are specific to NGTxCs and can assess the early key events of inflammation and the immune response. These are not necessarily solely focused on immune-related mechanisms but can identify them using transcriptomics. For instance, using transcriptomic tools, specific immune-related inflammation endpoints have been identified within in vitro cell transformation assay (CTA) models pertaining to later stages of proliferation and morphological changes leading to tumor formation [19]. For the BALB/c 3T3 CTA model, the most comprehensive set of endpoints was identified and included the classical and alternative complement pathways IFN, IL-1, IL-4, IL-6, IL-9, IL-17, IL-18, and TNFR2, as well as NF-κB. Additionally, the Bhas 42 CTA model has been shown to identify several key endpoints, including IL-1, IL-2, IL-6, and TNFR2, as reviewed in [19].

Among the mechanisms by which NGTxCs can induce cancer is by affecting the immune system, including, for example, the promotion of the differentiation of regulatory T cells (Tregs), which suppress the anti-tumor immune response and promote tumor growth [61]. Additionally, NGTxCs may inhibit the differentiation and function of effector T cells, which are responsible for killing tumor cells [62]. The dysregulation of the Aryl hydrocarbon Receptor (AhR) canonical pathway by environmental toxicants is marked by the altered expression of several cytokines and chemokines, including IL-22 and IL-17, leading to the activation of a disrupted inflammasome and of TME, supporting the epithelial–mesenchymal transition (EMT) and the progression to cell malignancy [19,63].

## 6. Key Molecular Players in Cancer Immune Evasion

There are a number of pivotal molecular players in cancer immune evasion, and an introduction to the roles they play is described in this section.

Beginning with the MIE of receptor activation, whilst the perturbation of hormone-related receptors plays a big role in immune dysfunction, it is the deregulation of the AhR pathway in particular that facilitates the degradation of the immune environment, enabling carcinogenic pathways to flourish, and for this reason, a more detailed evaluation is discussed. A number of key transcription pathways are also pivotal, principally Foxp3 and JAK-STAT, together with the microenvironment, TGF-β and AhR.

### 6.1. The Molecular Role of the Aryl Hydrocarbon Receptor

The AhR is a ligand-activated transcriptional factor belonging to the ancient superfamily Pern–Arnt–Sim, whose members facilitate communication between the host and environmental factors and are highly involved in development and metabolism. As such, it has long been considered an environmental sensor and a main target of the exposome.

Many environmental chemicals can act as agonist ligands of AhR, contributing to the dysregulation of the various AhR-regulated pathways and playing a role in immune evasion. Polycyclic aromatic hydrocarbons (PAHs), dioxins, and dioxin-like chemicals are all well-known AhR ligands.

The AhR is activated by planar dioxin-like ligands such as 2,3,7,8-tetrachlorodibenzo-p-dioxin (TCDD) and polychlorinated biphenyl (PCB) 126. Ligand binding induces conformational changes in the AhR ligand-binding domain (LBD), including π–π stacking interactions that stabilise the receptor–ligand complex [64,65]. In contrast, other ligands may engage the LBD via hydrogen bonding, as shown in homology docking studies (e.g., [66]). Heat shock proteins (HSPs), particularly HSP90, are critical for maintaining the correct conformational state of AhR during these transitions and were essential in the recently successful elucidation of its crystal structure [67]. These molecular interactions are essential for the AhR-mediated regulation of immune cell differentiation and immune responses. Ligand binding triggers AhR translocation into the nucleus, where it binds to xenobiotic response elements (XREs) in the DNA to initiate the transcription of target genes involved in detoxification, metabolism, and immunity [68,69]. Prior to nuclear translocation, ligand binding prompts the dissociation of AhR from cytosolic chaperones, including HSP90 and AhR-interacting protein (AIP), facilitating conformational rearrangement.

Once in the nucleus, AhR forms a heterodimer with the AhR nuclear translocator (ARNT). This complex binds to XREs in the promoter regions of target genes, leading to their transcriptional activation [70]. The canonical AhR signalling pathway involves the recruitment of transcriptional co-regulators. Co-activators such as steroid receptor coactivator-1 (SRC-1) and p300/CBP-associated factor (P/CAF) enhance transcription, whereas co-repressors like nuclear receptor co-repressor (NCoR) and silencing mediators of the retinoid and thyroid hormone receptor (SMRT) inhibit transcription by competing with co-activators for binding to the AhR–ARNT complex [71,72].

Nutrients such as indole carbinols from cruciferous vegetables are also ligands with a protective carcinogenic role [73] and were reviewed in Jacobs and Lewis in 2002 [65]. Chemicals that help restore a more balanced activity of AhR will have value when selecting proficiency chemicals for the AhR-relevant test method validation and inclusion in the NGTxC battery. It is now well established that AhR signalling plays a fundamental role in both innate and adaptive immune responses via several mechanisms (Figure 4).

The activation of tryptophan metabolism is considered to be one of the main events in the AhR-mediated immune response due to the ability of tryptophan metabolites, such as kynurenines, to act as ligands for AhR. The liver and peripheral lymphocytes sustain tryptophan metabolism through the production of indoleamine 2,3-dioxygenase (IDO1) (Figure 5).

AhR activation influences the differentiation and function of immune cells such as iTregs, Th17 cells, dendritic cells, and natural killer (NK) cells. Indeed, AhR activation in the presence of TGF-β promotes the generation of Foxp3+-iT(reg) cells [74]. Moreover, AhR can modulate the production of pro-inflammatory cytokines and chemokines, which play crucial roles in immune responses.

Th17 cells are a distinct subset of CD4+ T cells that secrete Th17 cytokines and are essential for the development of various inflammatory diseases. These cells are differentiated when exposed to IL-6 and TGF-β [75]. The differentiation of Th17 cells is regulated by the transcription factors ROR-α and RORγt, whilst Stat3 is critical for IL-17 production [76]. The aryl hydrocarbon receptor (AhR) supports Th17 cell development through multiple pathways. It can bind xenobiotic response elements (XREs) in the Th17 promoter region to regulate gene expression, and it also acts in concert with Stat3 to enhance the expression of Aiolos (IKZF3), a member of the Ikaros transcription factor family, thereby suppressing IL-2 and facilitating Th17 cell generation [77].

AhR exerts these functions either by directly binding to gene promoters or by modulating intracellular signalling pathways. AhR canonical signalling leads to the transcription of cytochrome P450 1A1 (CYP1A1), which is essential for the metabolism of many xenobiotics, including PAHs. CYP1A1 also plays a crucial role in the immune response and immune evasion by regulating xenobiotic detoxification, stimulating the production of IL-22 and antimicrobial peptides and mucus, and modulating AhR signalling. With respect to relevant in vitro tools for the measurement of CYP induction, including CYP 1A1 measurement, a test method has been validated [78], and the chemical applicability domain has been extended [79] with further validation work being conducted (manuscripts in preparation), which will be proposed for OECD TG adoption in the near future. A number of AhR in vitro test methods are available, of which one is also currently undergoing ISO validation [13], and further examples are identified in a recently published inventory relevant to thyroid test methods [80].

The AhR–RORγt–Foxp3 axis is also crucial in regulating the expression of the orphan chemoattractant receptor GPR15 in both mice and humans. This receptor facilitates the homing of T lymphocytes to the large intestine and contributes to the maintenance of intestinal immune balance and a healthy gut microbiota [81]. Mechanistically, AhR promotes Gpr15 expression by binding to accessible chromatin regions within its locus. The transcriptional activity of AhR in this context is influenced by Foxp3 and RORγt, which are both predominantly expressed by intestinal regulatory T cells (Tregs) in vivo. Foxp3 enhances AhR’s ability to bind DNA at the Gpr15 site, thereby upregulating GPR15. In contrast, RORγt appears to act as a negative regulator, at least in part by competing with AhR for access to the same genomic region [81].

The deregulation of the AhR pathway and its byproducts in the TME facilitates immune evasion, which in turn helps the tumor to evolve. Despite the absence of any significant AhR mutations in any cancer, high AhR activity in cancer is common because of the increased production of endogenous ligands and recruitment of exogenous ligands. This is also part of the AhR–ER interplay in estrogen-positive and -negative cancers, as seen, for example, in breast and ovarian cancers, and overall low AhR expression is associated with reduced cancer risk [82,83].

### 6.2. Transcription Factor Foxp3

Treg cell function and phenotype are governed by the transcription factor Foxp3. Foxp3 was first identified as the gene responsible for severe autoimmune and inflammatory syndromes, including immune dysregulation, polyendocrinopathy, enteropathy, and X-linked (IPEX) syndrome [84]. Since then, numerous studies have elucidated the molecular mechanisms underlying the function of Foxp3.

Foxp3 acts as a transcription factor by binding to specific DNA sequences, thereby influencing the expression of target genes [85] (Figure 3). Several studies have identified the DNA-binding sites of Foxp3 and its associated cofactors, providing insights into the regulatory networks controlled by this transcription factor. Additionally, post-translational modifications of Foxp3, such as phosphorylation and acetylation, have been shown to affect its stability and activity [86].

Foxp3 is essential for the Treg-suppression function of these cells. It controls the expression of genes involved in immune suppression, including interleukin-2 receptor alpha (IL-2RA) and cytotoxic T lymphocyte-associated protein 4 (CTLA-4) [87]. Foxp3 also interacts with other transcription factors such as NF-AT, RUNX1, and STAT5 to regulate Treg-specific gene expression and ensure Treg phenotype stability [87].

The importance of Foxp3 in immune regulation is further highlighted by its dysregulation. Mutations in the Foxp3 gene result in the loss of Treg function and the development of autoimmunity, as observed in IPEX syndrome [88]. Moreover, alterations in Foxp3 expression or function have been implicated in various autoimmune diseases and cancers [87].

However, Foxp3 alone is not sufficient to confer and maintain Treg cell function and phenotypes. Treg cell-specific epigenetic changes are critical in Treg cell specification, controlling its potential plasticity, and establishing a stable lineage [89]. IL-27’s stimulation of Treg cells induces the expression of Lag3, a surface molecule implicated in negatively regulating immune responses. Lag3 expression in Tregs is critical for mediating Treg function in suppressing colitogenic responses. Furthermore, Lag3 supports Treg cell function by limiting the expression of c-Myc, a key regulator of aerobic glycolysis. Moreover, in addition to the immunosuppressive role of Tregs, these cells have non-suppressive roles, including the induction of anti-inflammatory macrophages, the inhibition of foam cell formation, the influence on cholesterol metabolism, and suppression of immune responses of endothelial cells and innate lymphoid cells [87].

Tregs play a critical role in maintaining immune homeostasis and preventing autoimmunity. However, in the TME, Tregs can suppress effector T cell function and limit anti-tumor immune responses [87].

### 6.3. Cytokines

As previously discussed, cytokines play a fundamental role in controlling immune responses. Therefore, they can be regarded as useful molecular targets to distinguish between immune surveillance, where they stimulate anti-tumor responses, and immune evasion in cancer, where tumor cells exploit cytokine signalling to evade immune detection and promote tumor growth.

Cytokines are small molecules with masses ranging from 8000 to 40,000 daltons and include chemokines, tumor necrosis factors, haematopoietic growth factors, and interferons [90]. As noted in Section 2, the classification of cytokines is a challenging task because of the continuously evolving understanding of their diverse functions and roles within the immune system [22]. As new discoveries emerge, traditional classifications based on specific criteria, such as structure, receptor binding, or cellular sources, may need to be revised or expanded to accommodate the growing complexity of cytokine biology. Additionally, cytokines often exhibit pleiotropic and redundant activities, meaning that a single cytokine can have multiple effects on different cell types and biological processes, further complicating classification. Despite these challenges, one common approach to classifying cytokines is based on their roles in inflammation, which broadly categorises cytokines as pro-inflammatory, anti-inflammatory, regulatory, or adaptive [22]. Pro-inflammatory cytokines, such as tumor necrosis factor-alpha (TNF-α), interleukin-1 (IL-1), and interleukin-6 (IL-6), promote inflammation and immune responses, whereas anti-inflammatory cytokines, including interleukin-10 (IL-10) and transforming growth factor-beta (TGF-β), help regulate and resolve inflammation. Regulatory cytokines such as interleukin-35 (IL-35) and interleukin-37 (IL-37), modulate immune responses and maintain immune homeostasis. Adaptive cytokines are involved in regulating adaptive immune responses, including T-cell differentiation and activation, B-cell function, and antibody production. Examples include interleukin-2 (IL-2), interleukin-4 (IL-4), interleukin-17 (IL-17), and interferon gamma (IFN-γ). IL-32 is a cytokine restricted to higher mammals, which fine-tunes multiple pathways involved in metabolic processes and host defence. It has been shown to promote breast cancer cell invasion and metastasis through the integrin β3–p38 MAPK signalling pathway (see Table 2).

This classification provides a framework for understanding the dynamic interplay between cytokines and immune processes; however, it is important to recognise that cytokine functions are context-dependent and can vary depending on the specific microenvironment and immune context.

The most striking example is represented by IL-6. IL-6 plays a dual role in immune regulation and in cancer progression. It has a crucial role in mediating acute immune responses and tissue repair during infection or injury. It stimulates the production of acute-phase proteins and mobilises immune cells to sites of inflammation, thereby contributing to host defence mechanisms. However, persistently elevated IL-6 levels are associated with chronic inflammation, autoimmune diseases, and cancer progression. In cancer development, IL-6 promotes tumor growth, metastasis, and immune evasion by activating oncogenic signalling pathways, enhancing cancer cell proliferation, survival, and angiogenesis and suppressing anti-tumor immune responses [91]. The complex interplay between IL-6 and the immune system highlights its context-dependent effects and underscores its potential as both a therapeutic target and biomarker for monitoring the progression of cancer and inflammatory disorders, but it also acts as an immune–inflammatory-related toxicity-predictive biomarker tool in the NGTxC IATA. Other cytokines have been recognised as promising tools in immuno-oncology, as they can stimulate anti-tumor immune responses, enhance immune cell function, and promote the activation and expansion of effector immune cells capable of targeting and eliminating cancer cells. For example, IL-2 and interferon-alpha (IFN-α) have been used in immunotherapies for certain types of cancer, particularly melanoma and renal cell carcinoma, to boost immune responses and induce tumor regression [92,93,94,95]. Additionally, cytokines, such as interleukin-12 (IL-12) and interleukin-15 (IL-15), are being investigated for their potential as cancer immunotherapies because of their ability to enhance anti-tumor immunity and promote the proliferation and activation of immune effector cells [96]. Similarly, these all have the realistic potential to be suitable immune-related toxicity biomarkers in the NGTxC IATA, as for IL-6.

By influencing diverse cellular processes at different stages of cancer progression, cytokines serve as critical markers that provide valuable insights into the dynamic nature of tumor development and metastasis. Indeed, pro-inflammatory cytokines, such as the IL-1 family, IL-6, and TNF-α, are key players in creating chronic inflammation in the tumor context [97]. Depending on the cancer type, TME, and other related factors, these cytokines have been shown to have both pro-tumorigenic and anti-tumorigenic functions. In terms of pro-tumorigenic functions, pro-inflammatory cytokines, along with supportive cells and factors, lead to hyperinflammation and tumor promotion by affecting various mechanisms, including angiogenesis, proliferation, immunosuppression, extracellular matrix remodelling, invasiveness, and metastasis.

In Figure 6, we present the progression of colon cancer characterised by distinct stages, each marked by specific cytokine profiles categorised into pro-tumorigenic and anti-tumorigenic groups. These cytokines serve as critical markers reflecting the dynamic interplay between inflammation and tumor progression in colon cancer. During cancer progression from dysplasia to invasive carcinoma, the cytokine profile undergoes dynamic shifts: initially, pro-tumorigenic cytokines support tumor growth and immune evasion; subsequently, anti-tumorigenic immune responses, mainly mediated through immunosuppression, are activated; finally, immune dysfunction re-establishes a pro-tumorigenic microenvironment that facilitates tumor invasion and metastasis.

### 6.4. Transforming Growth Factor-β: The Master Cytokine

One of the main mechanisms by which Tregs suppress immune responses is the production of TGF-β, a cytokine with potent immunomodulatory properties. Accumulating evidence suggests that TGF-β plays a key role in maintaining immune homeostasis and fostering tumor immune escape. TGF-β controls the innate immune response by orchestrating the performance of macrophages and neutrophils and creating a network of negative regulatory signals. It controls adaptive immunity by directly promoting the expansion of naturally occurring Tregs. It can also induce apoptotic signals in premalignant cells.

Tumor cell macrophages with an M2 phenotype produce high levels of active TGF-β, aiding the conversion of CD4 T cells into suppressive Tregs, which ultimately promotes TGF-β-driven EMT and immune evasion. In addition, M2 cells produce IL-10 and VEGF, together with IL-1 and IL-6. IL-8, a colony-stimulating factor, significantly contributes to tumor growth.

When the TGF-β signalling pathway remains functional, cancer cells may bypass its pro-apoptotic effects by uncoupling epithelial-to-mesenchymal transition (EMT) from apoptosis. This mechanism allows them to exploit EMT for tumor-promoting purposes. Beyond its role in driving tumor invasion and metastatic spread, TGF-β signalling also triggers gene expression programs that enhance the ability of cancer cells to penetrate and establish themselves in specific target organs [98]. Therefore, TGF-β seems to be the dominant master cytokine in the TME, playing a pivotal role in determining the fate of dormant tumors (Figure 4).

### 6.5. JAK-STAT Axis

Janus kinase signal transducer and activator of transcription (JAK-STAT) signalling plays a critical role in the regulation of immune processes, including tumor cell recognition and immune escape [99]. The Janus kinase (JAK) family consists of four non-receptor tyrosine kinases, JAK1, JAK2, JAK3, and TYK2, which are expressed differently across various cell types. Each member of this family plays specific roles in cytokine signalling and regulates distinct cellular processes. JAK proteins phosphorylate and activate the signal transducer and activator of transcription (STAT) proteins in response to cytokine binding. This activation allows STAT proteins to translocate to the nucleus and regulate the transcription of genes involved in cell growth, differentiation, and immune response.

STAT1 and STAT2, in turn, induce the production of type I and II interferons (IFNs) and their downstream pathways.

In contrast, STAT3 has been linked to cancer cell survival, immunosuppression, and chronic inflammation in the TME. It is activated by various cytokines, including members of the IL-6 family, the IL-10 family, and others such as IL-21, IL-27, G-CSF, leptin, and IFN-Is.

Unlike other STAT proteins that are more specialised, STAT3 has a widespread role across many cell types and is crucial in mediating responses to various cytokines and growth factors. Primarily, it negatively regulates immune responses, cellular growth, differentiation, apoptosis, tumor development, and metastasis. STAT3 modulates the differentiation of Th17 cells through various signal transduction stages, such as membrane binding, phosphorylation, and nuclear translocation. In addition, it enhances the immunosuppressive capabilities of tumor-associated macrophages and myeloid-derived suppressor cells. STAT3 overactivation is associated with immunosuppression and cellular transformation. Regarding the regulation of cell growth, differentiation, and apoptosis, blocking JAK, which relays IL-6 signals to STAT3, can suppress the expression of the anti-apoptotic protein Bcl-xl, leading to apoptosis [100].

The continuous activation of STAT3 is linked to the development of various cancers, including head and neck, breast, non-small cell lung, colorectal, and haematological malignancies. Recently, STAT3 was reported as a marker of prognostic and therapeutic value in kidney disease [101]. Additionally, high levels of STAT3 and IL-6 correlate with reduced chemotherapy effectiveness in patients with triple-negative or high-grade breast cancer.

STAT4 is crucial for the differentiation and growth of Th1 and helper follicular T (Tfh) cells. In addition, STAT4 enhances germinal centre response following viral infection. The transcription factor STAT5 consists of two isoforms, STAT5a and STAT5b, which share 95% sequence similarity and perform overlapping functions in the lymphoid system [100]. STAT5 is critically targeted by various cytokine receptors that are integral to lymphocyte development, particularly IL-2 and IL-7 [100]. STAT6 primarily facilitates the transmission of signals from IL-4 and IL-13. IL-4 triggers the activation of STAT6, which is critical for the differentiation of Th2 cells and the conversion of immunoglobulin isotypes.

The identification of cross-regulatory interactions, post-translational modifications, and non-canonical pathways has revealed greater complexity in how JAK-STAT signalling governs tumor initiation and progression. Research exploring the role of this pathway in anti-tumor immunity has uncovered a diverse array of downstream targets—including oncogenes, microRNAs, and various co-regulators—that influence specific cellular phenotypes in response to JAK-STAT activation. Despite this knowledge, efforts to develop therapies addressing the frequent dysregulation of JAK-STAT signalling in cancer have been limited by undesirable off-target effects.

#### The JAK–STAT Axis’s Interactions with AhR

The interaction between AhR and the JAK/STAT signalling pathway plays a critical role in regulating immune responses, particularly through the modulation of tryptophan metabolism and the production of immunosuppressive metabolites such as kynurenine. Both in vivo and in vitro studies have demonstrated that polycyclic aromatic hydrocarbons (PAHs), acting as AhR agonists, or β-naphthoflavone, an AhR inhibitor, can influence the activation of the JAK/STAT pathway by modulating the canonical AhR signalling pathway [63,102,103]. When activated by cytokines such as IFN-γ, the JAK/STAT pathway induces the expression of IDO, a key enzyme in the tryptophan catabolism pathway. IDO converts tryptophan into kynurenine, which then activates AhR. This interaction forms a feedback loop where AhR activation further influences the JAK/STAT pathway and modulates immune responses. This integrated signalling mechanism is illustrated schematically in Figure 7. This pathway is particularly significant in the context of cancer development, where the upregulation of IDO and the subsequent increase in kynurenine levels create an immunosuppressive environment that aids tumor immune evasion [104]. Both JAKs and STATs are potential therapeutic targets. JAK inhibitors (JAKinhibs) include several marketed drugs used to treat a range of inflammatory and autoimmune conditions as well as certain types of cancer [105].

In contrast, the inability of selective IDO1 inhibitors to effectively inhibit tumor growth in clinical trials highlights the complexity of this pathway and suggests that broader strategies targeting both IDO activity and its interactions with the JAK/STAT and AhR pathways may be necessary for more effective cancer therapies [106].

## 7. Status and New Perspectives on the Use of Molecular Targets to Study Cancer Immune Modulation Using Alternative Methods

Currently, the assessment of chemical immune modulation and dysfunction predominantly relies upon animal models and assays designed to evaluate immunosuppression and immunotoxicity. However, there is a growing regulatory recognition of the importance of developing alternative methods, particularly in vitro approaches, to reduce reliance on animal testing, enhance efficiency, and provide more mechanistic insights into the immunomodulatory effects of chemicals. With respect to OECD TG development and adoption, this is now well developed and recently adopted for immunotoxicity in relation to IL2; the first in vitro IL2 TG was published in 2023 [58]. In addition, immune-related endpoints such as skin sensitisation have in vitro TGs to address relevant Key Events (e.g., [59]).

The following section summarises the current status of regulatory research efforts and the regulatory-relevant immune dysfunction sources of information that can support the development of and transition to alternative methods for assessing chemical immunotoxicity. The focus is on the advancements in in vitro approaches and the understanding of relevant human biopsy analyses for their practical potential to complement or replace traditional animal models and assays. This is first underpinned by a consideration of the importance of the evolutionary conservation of key immune signalling mechanisms and their frequent duality and species specificity.

### 7.1. Consideration of the Importance of the Evolutionary Conservation of Key Immune Signalling Mechanisms

The evolutionary conservation of key immune signalling molecules provides an invaluable framework for studying immune modulation in cancer across diverse species, enabling researchers to utilise alternative models to uncover fundamental mechanisms that are relevant across vertebrates, including humans. This conservation allows insights from non-mammalian systems to be applied to human cancer immunology (and vice versa), offering new avenues for understanding and targeting immune pathways in cancer therapy.

The immune system and pathogens are engaged in an evolutionary arms race, where each constantly adapts to the changes in the other [107]. This co-evolutionary pressure forces signalling molecules like cytokines to undergo continual change. Over time, ILs have evolved to fine-tune the immune response, balancing between effective immune defence and the prevention of excessive inflammation that could damage host tissues [108]. Specific ILs, such as IL-10, which has both pro-inflammatory and anti-inflammatory functions, have undergone significant evolution in different species, reflecting the diverse challenges posed by pathogens in various environments.

Since its discovery in 1989, IL-10 has been thoroughly investigated for its potent anti-inflammatory and regulatory functions in immune responses associated with infection and disease, and this has shed new light on the cross-regulation between T helper 1 (Th1) and T helper 2 (Th2) cytokines [109]. The first non-mammalian IL-10 sequence was found in teleost fish in 2003 [110], with chicken IL-10 identified the following year [111]. Interestingly, some fish species exhibit gene duplications of IL-10Rs, which adds intricacy to their immune regulation, potentially allowing for more refined control mechanisms. Gene duplication in teleost fish does not necessarily equate to functional redundancy, suggesting that these additional gene copies may provide more precise regulation of IL-10 activities [112].

Besides IL-10, other interleukins such as IL-17 have been identified throughout the phylogenetic tree, highlighting their evolutionary conservation across various species. IL-17, originally characterised in mammals, has homologs that extend to non-mammalian species like teleost fish. In mammals, six IL-17 members (IL-17A, B, C, D, and F) have been identified, with IL-17A and IL-17F showing the highest similarity, whilst IL-17E (IL-25) is the most divergent. Teleosts not only share mammalian homologs such as IL-17A/F1, IL-17B, and IL-17C, but they have also evolved a unique ligand, IL-17N, specific to their lineage. This illustrates the adaptive evolution of the IL-17 cytokine family, allowing species-specific expansions and functional diversifications, especially in immune regulation and host–microbe interactions in the gut [113].

#### Mammalian Models

Immunotoxicity testing in in vivo mammalian models targets several of the key molecular players identified here and has been used to highlight chemical toxicity in rodents, in particular for agrochemicals; however, species specificity can vary. However, the T-cell-dependent antibody response (TDAR) immunosuppression method has remained core to immunosuppression regulatory testing and assessment to date.

The interspecies difference between mouse and human IL-10 production begins with differences in thymic development and function. In mice, thymic involution occurs rapidly after sexual maturation (around 6 weeks of age), leading to a reduced, yet persisting, production of new T cells throughout life. In contrast, in humans, thymic involution is more gradual, with significant thymic activity persisting into early adulthood and contributing to the maintenance of a more diverse peripheral T cell repertoire [114].

Thymic involution results in a decline in the production of new naïve T cells, leading to a less diverse T cell receptor (TCR) repertoire and impaired immune responses in both humans and mice. This decline can potentially impact IL-10 T cell populations, including both classical FoxP3 regulatory T cells (thymus-derived Tregs) and FoxP3 Tr1 cells (induced in the periphery). Many IL-10-producing cells, such as regulatory T cells and effector T cell subsets (e.g., Th1, Th2), are initially derived from naïve T cells that exit the thymus and differentiate in peripheral tissues. As thymic involution reduces the influx of naïve T cells, the available precursors for these IL-10-producing populations also decline, potentially leading to a lower overall IL-10 production capacity in the immune system [115,116].

In mice, IL-10 production is primarily associated with Th1 cells within the thymus, whereas in humans, both Th1 and Th2 cells contribute to IL-10 production [117]. This broader production profile indicates that IL-10 plays a more expansive role in modulating immune responses. In humans, IL-10 is crucial in both Th1-mediated and Th2-mediated immune responses, helping to balance pro-inflammatory activities and prevent immune pathology in a wider range of immune environments. For example, Th2 cells contribute to IL-10 production in contexts such as allergic responses and helminth infections, where the Th2-type immune response is dominant [118].

This distinction reflects broader differences in how IL-10 regulates immune responses in both species, particularly in shaping the balance between pro-inflammatory and anti-inflammatory signals during T cell development. Understanding these differences is crucial for translating findings from mouse models to human applications, especially in immunology and cancer research.

A further regulatory tool that is currently in use is the OECD TG 443 Extended One-Generation Reproductive Toxicity (EOGRT) test method [52]. This reproductive and developmental-focused in vivo TG assessing pre- and postnatal chemical exposure also has a dedicated developmental immune effect offspring (weanling) cohort. Two weeks prior to mating and during the mating, gestation, and weaning periods, the parent animals are exposed to graduated doses of the test substance.

At weaning, the F1 offspring pups are selected and assigned to cohorts of animals for reproductive/developmental toxicity testing (cohort 1), developmental neurotoxicity testing (cohort 2), and developmental immunotoxicity testing (cohort 3). The F1 offspring receive further treatment with the test substance from weaning to adulthood. Clinical observations and pathology examinations are performed on all animals for signs of toxicity, with special emphasis on the integrity and performance of the male and female reproductive systems and the health, growth, development, and function of the offspring. Part of cohort 1 (cohort 1B) may be extended to include an F2 generation; in this case, procedures for F1 animals will be similar to those for P animals. Whilst the assessment of immune-related dysfunction is not the initial objective of this test method, when requested, this test method’s cohort 3 is designed to provide pertinent developmental immune-relevant information. This can be utilised within the weight of evidence assessment in the IATA for NGTxC.

### 7.2. Relevance of AhR and Non-Mammalian Vertebrate Models

The basic structure and function of AhR are highly conserved across vertebrate species, including fishes and mammals. This suggests that the mechanisms underlying AhR activation and its effects on immune regulation are likely to be similar in fish and humans. In many cases, fish-based assays are recognised as good surrogate approaches for detecting human toxicity, including neurotoxicity [119] and immunotoxicity [120] but also vice versa, human relevant test methods are a good starting point for assessing aquatic immune dysfunction, also in relation to e.g., fish NGTxC mechanisms, in the absence of species-specific methods, or where the performance of the test methods cannot separate immunotoxicity adequately [121,122].

### 7.3. In Vitro Immunotoxicology Tools

As indicated in the introduction to Section 7 above, whilst, to date, the majority of successes in using in vitro models to assess immunotoxicity have focused on chemical sensitisation, and in particular, skin sensitisation [123,124], and IL2 [58] in recent decades, there has been progress in identifying immunosuppressive chemicals [55,61,62,125]. Current standard assessments of immunotoxicity rely on vitro and ex vivo assays that evaluate different functional parameters of the immune response, among which the cytotoxic T-lymphocyte (CTL) assay and the natural killer (NK) cell activity assay are relevant for the immunosurveillance of cancer [12] and should be further explored for the in vitro identification of chemical-induced immunosuppression. These assays involve the in vitro stimulation of lymphocytes and do not require the use of animals specifically sensitised with antigens, which are easily applicable to immune function assessment and for characterising in vitro the direct action of immunotoxicity upon their addition to cultured lymphocytes.

Whilst testing chemicals using ex vivo/in vitro immune functional tests will provide information on their ability to deregulate an immune response, the investigation of the ability of chemicals to modulate the TME and its interactions with the tumor and immune cells will require different models and tools and the application of transcriptomics in particular. In the development and progression of solid tumors, it is well known that the tumor and surrounding microenvironment are in constant communication and evolve together, selecting for traits that favour tumor growth and invasion, which are tumor-specific [19,126]. In this regard, a recent systematic literature review of publications on immuno-oncology describing promising advanced model developments published by the JRC’s EU Reference Laboratory for alternatives to animal testing (EURL ECVAM) provides a list of 542 studies (literature from 2014 to 2019) reporting human biopsy data and in vitro models used in the immune-oncology field for therapeutic applications [127]. Among the therapeutic designs of in vitro models identified, the majority were colorectal, breast, melanoma, pancreatic, ovarian, and non-small-cell-lung-cancer models [127]. These models undoubtedly offer the possibility to study biological mechanisms by which NGTxCs may favour cancer progression and immune evasion and may be of a higher level of relevance for entry into the NGTxC IATA once critically evaluated.

Therefore, preliminary filtering and exploration of these therapeutic tools (listed in the supplementary files of Dura et al. [127]) were conducted to assess potential applications towards the screening of chemicals that can perturb the immune pathways that trigger NGTxC. Specifically focusing on well-controlled cancer biopsy information, human-relevant cancer biomarker information was extracted and further assessed for pivotal/significant immune-oncology-related biomarkers that can be used to support the evidence base and can be applied to the current NGTxC IATA test method toolkit. These biomarkers can also have the potential to be identified in the transcriptomics versions of the CTAs [19], which address the KE of cell transformation in the IATA, as well as potentially in 28 and 90 in vivo studies. The biomarkers and references are provided in Table 3. The complexity of the in vitro approach depends upon the possibility of identifying a common thread that can unite immunosurveillance to all tumors or, on the other hand, whether more representative models of the different types of tumors are needed. From Table 3, we can see that there are commonalities in the biomarkers being observed in human biopsies, in particular in relation to PD-L1, MAPK, mTOR, TIL and Tregs, CD8+, Fox p3+, WNT, and IL-11.

Indeed, as already identified herein, together with IL-17, IL-11 is highly upregulated in many types of cancers and is one of the most pivotal cytokines during tumorigenesis (and metastasis).

#### 7.3.1. In Vitro Methods to Evaluate Cytokine(s)

Given the importance of cytokines in immune regulation, methods for measuring cytokine levels have been developed for therapeutic uses to allow researchers and clinicians to quantitatively analyze cytokine expression patterns, monitor immune responses, and evaluate the efficacy of cytokine-based therapies in various disease settings, including infectious diseases, autoimmune disorders, and cancer (Table 3). These approaches also have great utility to be used in combination with the CTA in the NGTxC IATA and are commercially available.

Quantifying cytokines offers considerable value in both clinical medicine and biology, as they provide insight into physiological and pathological processes, which can aid in highlighting the response to chemical exposure and therapeutic treatments. Whilst their role in immune responses is undoubtedly central, the toxicological significance of their modification is difficult to interpret as they are intermediate parameters. Their measurements should therefore be combined with the assessment of apical parameters to allow easier interpretation of immunotoxic effects.

In vitro cytokine detection is a versatile and reliable method widely adopted by the research community for mechanistic understanding. A broad range of samples, including primary cells, tissues, and bodily fluids, have been used for in vitro cytokine testing. Several techniques, including enzyme-linked immunosorbent assays (ELISAs), polymerase chain reactions (PCRs), and advanced biosensors, including point-of-care testing, have been developed. Noting a precedent in the context of regulatory toxicology applications, which is that some of these methods have been adopted as TGs [58,128,129].

**Table 3 ijms-26-06310-t003:** Key immuno-oncology biomarkers identified from human cancer biopsy studies.

Tissue	Biomarker	Study Design	Results	Relevance for NGTxC IATA Immune Endpoint Utility, Considerations on How to Apply and Interpret	Reference(s)
Breast cancer (ER+ and ER−)	IDO1, DNA methylation profile	- Sera and biopsy tissue from ER+ and ER− breast cancer patients. - Methods: ELISA for kynurenine detection, HPLC, Western blot, DNA methylation by pyrosequencing and MassARRAY analysis, luciferase reporter assays. - Kinetic modelling using Trp concentration (5 µM median of healthy donors).	- Serum Kyn and tumor IDO1 expression lower in ER+ vs. ER− patients. - Confirmed in larger datasets: lower IDO1 mRNA and protein expression in ER+ tumors. - Statistical significance: *p* < 0.01 and *p* < 0.001.	- High relevance: IDO1/TDO2 axis mediates immune suppression. - Supported by evidence from JAK-STAT pathway studies.	[128]
Renal Cell Carcinoma (RCC)	TILs PD-1/PD-L1	- Matched primary and metastatic RCC samples. - Analysis of CD8+, CD4+, and Foxp3+ TILs; PD-1/PD-L1 status.	- Metastases show lower CD8+; Foxp3+ ratios, higher PD-L1 expression. - Primary tumors have higher CD8:Foxp3 ratios.	- High relevance: High PD-L1 and Foxp3 associated with immune escape and tumor progression.	[130]
Craniopharyngioma (Brain tumors)	PD-1/PD-L1 checkpoint pathway	- Snap-frozen ACP and PCP samples. - Transcriptomic profiling (Affymetrix arrays) compared to other pediatric tumors and normal brain.	- Strong PD-L1 expression in ACP and PCP. - PD-1 intrinsic expression with MAPK/mTOR pathway activation in ACP.	- High relevance: confirmed immune evasion pathways, supported by melanoma metastasis models.	[131]
Colorectal Carcinoma (CRC)	TILs, Th1, Th2, Th17, Treg markers	- TIL counts in 199 CRC samples. - Markers: RORγT (Th17) and Foxp3 (Treg) ratios to CD3.	- High RORγT/CD3 linked to lymph node metastasis (*p* = 0.002). - Foxp3/CD3 ratio linked to tumor localisation. - RORγT/CD3 ratio: independent prognostic marker (*p* = 0.04; HR 1.84).	- High relevance: Key indicators of tumor immunity vs. immune suppression balance.	[132]
Gastric Cancer	CD8+ T cells and PD-L1	- 147 patients (35 EBV+, 112 EBV-). - Stratified analysis for immune markers.	- High CD8+ and PD-L1 expression linked to the worst prognosis (*p* = 0.015). - EBV infection linked with increased CD3+ T cells.	- High relevance: PD-L1 and EBV status critical in prognosis and immune environment modulation.	[133]
Colorectal Carcinoma	IL-11 and phosphorylated STAT3	- Tumor tissues from 7 colorectal cancer patients. - Analysis of IL-11 expression and p-STAT3 activation.	- Increased IL-11 mRNA and protein in tumors. - Phosphorylated STAT3 correlates with MDSC differentiation markers (CD11b+ CD14+).	- High relevance: IL-11/p-STAT3 axis critical for shaping immunosuppressive tumor microenvironment.	[134]

The IL-2 Luc assay is technically relatively simple, with a short duration, and validation has demonstrated reliable results based upon a well-understood immune mechanism. It utilises the Jurkat-derived IL-2 reporter cell line 2H4. This cell line contains luciferase genes controlled by the IL-2 and GAPDH promoters, allowing for the quantitative measurement of luciferase gene induction in response to immunotoxic chemicals. Specifically, the IL-2 promoter-driven luciferase activity (IL2LA) and GAPDH-promoter-driven luciferase activity are measured to assess immunotoxic effects, based on observations showing similar suppressive effects of chemicals on IL2LA and IFN promoter-driven luciferase activity (IFNLA). Therefore, the IL2LA is used here as the primary indicator of immunotoxicity, and it is a valuable method for assessing chemicals affecting T cells. However, it is insufficient on its own to derive immune dysfunction-related conclusions by the regulator and must be used as part of a testing battery.

Furthermore, whilst this assay is suitable for soluble test chemicals or those that form stable dispersions, it shares common limitations with many suspension-cell-based assays, particularly when testing highly hydrophobic substances. In addition, the interference of luciferase activity by test chemicals can lead to misleading results, such as apparent inhibition or increased luminescence. Another limitation of the IL-2 Luc assay involves its focus on cellular response mechanisms downstream of phospholipase C activation, a mechanism by which chemicals such as phorbol esters can trigger cell proliferation, thereby acting as NGTxCs. As a result, this assay may not detect chemicals that target upstream signalling molecules or other signalling pathways, and a more comprehensive collective cytokine disruption assessment is needed to address immune dysfunction in relation to NGTxC.

#### 7.3.2. Considerations Towards a Practical Approach for the Comprehensive Hazard Assessment of Immune Dysfunction for the NGTxC IATA

In the preceding sections of this paper, we have summarised the regulatory approaches being taken to assess immunotoxicity historically and looked at their applicability for the NGTxC IATA, in particular with respect to the evolution of cytokines and the yin yang balancing mechanisms of key cytokines that need to be included to ascertain immune dysfunction in relation to carcinogenicity. Whilst there are very few in vitro tools for specific cytokines adopted as TGs, in relation to, e.g., skin sensitisation that could have applications for the NGTxC IATA, the question that arises is; Is it really practical to try to develop a whole series of in vitro IL test methods relevant for NGTxC? For this would require extensive funding for validation, the data interpretation procedures would be intricate and complex, and the time scales would be long. Furthermore, intellectual property issues would be likely, and the licensing and operational fees for such a large battery of in vitro tests intended for regulatory purposes would not be insignificant in practice for industry end-users.

A more practical, scientifically relevant and economical solution would be to instead derive dedicated, endpoint-customised cassettes of pivotal KE biomarkers, including cytokine markers, as identified in Table 3 and Table 4, as ready-to-use kits, where the counterbalancing roles and activities of the cytokines are incorporated and they are optimally combined to address specific target endpoints. These could then have applications for both clinical diagnostic purposes, as well as within the NGTxC IATA and the weight of evidence assessment, according to the modular approach, as indicated in Table 5 with descriptions of the biological mechanisms that can be assessed and implications to support the hazard interpretation, summarised in Table 6.

## 8. Discussion and Recommendations for Immune Dysfunction Relevant Tool Development for the OECD NGTxC IATA

Understanding the role of immunotoxicity and immune evasion induced by NGTxCs is pivotal for the development of a robust IATA. NGTxCs can subtly affect immune function and contribute to cancer progression through mechanisms of immune modulation.

Immune modulation encompasses both immunosuppression and immunoenhancement, which can significantly affect the safety profiles of chemicals, including non-genotoxic carcinogens. A thorough understanding of immune modulation mechanisms is vital for developing robust testing and assessment strategies and ensuring accurate risk evaluation and management. This understanding is fundamental for advancing regulatory frameworks, improving the safety assessment of new substances, and protecting public health.

### 8.1. Recent International Pharma Updates in Relation to Immune Dysfunction and NGTxC

International efforts are underway to develop testing strategies aimed at comprehensively understanding the potential carcinogenicity of various chemicals, but especially pharmaceuticals. The International Council on Harmonisation (ICH) S1B(R1) carcinogenicity global testing guideline for pharmaceuticals has been very recently updated with an addendum introducing a comprehensive integrated Weight of Evidence (WoE) approach to determine the necessity of a 2-year rat carcinogenicity study [170]. The WoE integrated assessment involves evaluating the immune modulation factor according to the ICH S8 guidelines; the latter are applicable to new human pharmaceuticals [54]. These guidelines limit immunotoxicity to ‘unintended immunosuppression and immunoenhancement,’ excluding allergenicity or drug-specific autoimmunity’, and apparently, there are no plans to update the guideline in the near future. The TDAR assay remains core to both pharma and industrial chemical immune-toxicological testing.

The new FDA guidance on the Nonclinical Evaluation of the Immunotoxic Potential of Pharmaceuticals [171] adds details on assessing immune function related to carcinogenicity, emphasising the need to consider a pharmaceutical candidate’s potential to promote, grow, and metastasise tumors. It also highlights the importance of evaluating the effects of key immune components involved in tumor surveillance (e.g., NK cells, T cells, and antigen-presenting cells), such as the downregulation or functional impairment of critical immune cell populations [171]. Intriguing insights into pharmaceuticals have shed light on a fresh perspective regarding the cancer risk associated with immune modulation. In the development of pharmaceutical carcinogenic therapeutics, prioritising the crafting of testing strategies that account for immunosuppression is imperative. Although not universally applicable, many cancers associated with chronic immunosuppression, such as those observed in transplant recipients or individuals with HIV/AIDS, are linked to chronic infections, such as viruses, bacteria, or parasites [172]. These mechanisms may occur simultaneously [173]. Similar mechanisms can be attributed to the concurrent presence of chemicals and viruses, both of which target the same molecules involved in immune-mediated inflammation [107]. As already noted, it is generally acknowledged that the 2-year rodent carcinogenesis bioassay (RCB) is not considered sufficiently informative to highlight the multifactorial nature of the complex link between immunosuppression and cancer risk. This is especially evident when the tumor hazard stems from the reactivation of latent viral oncogenes, infectious agents, or chronic inflammatory conditions. The US FDA have recently acknowledged that these factors exhibit significant species variation, posing challenges for human translation [171] (as also discussed in Paparella et al. [174]).

Learnings from regulatory developments in pharma can be applied to the regulatory industrial chemical space, building upon a thorough understanding of immune modulation mechanisms beyond ‘immunotoxicity’. For example, AOP thinking is also being applied by the Innovative Medicine Initiative to evolve immunomodulatory safety assessment for new therapeutic drugs and clinical assessment. For these purposes, they consider that the improved assessment of chimeric antigen receptor T cells (CAR T cells), already core to the immune checkpoint inhibitors (ICIs), and cytokines (e.g.,IL-2), is needed to support the assessment of immunomodulatory drugs. This is vital for developing robust testing and assessment strategies and ensuring accurate hazard evaluation and ultimately risk management.

### 8.2. Building Immune Dysfunction Tools into the NGTxC IATA

Looking at the KEs in the OECD NGTxC IATA and the tools and approaches already identified and published in the Special Issue on NGTxC mechanisms, pivotal immune-related markers in relation to Module A of the existing information on the modular strategy for the testing of NGTxCs [17] and cancer databases have already been identified [57], as well as in relation to the KE of immune dysfunction, herein. For the later KE of cell transformation, the CTAs are able to highlight the cytokine and other relevant markers [19], and commercial kits, also including ELISA, are readily available for specific cytokines.

Immune-related biomarkers in the NGTxC IATA are shown in Figure 8A in the lower pink blocks, and then, in Figure 8B, specific biomarkers for each MIE/KE are further organised according to the flow of the NGTxC IATA.

However, we also propose that separate batteries of key immune biomarkers be addressed:(1)Perturbation of inflammatory markers;(2)Anti-inflammatory markers;(3)Adaptive immunity markers, as summarised in Table 4, to be selectively and optimally developed in kit form.

These could be used within an initial screening and be designed to be relevant for each module of the NGTxC IATA and the relevant KE, as well as the forthcoming proposed decision tree approach from the OECD Expert Group (OECD June 2025). As discussed in Section 7.3.2, this might be a way to achieve more rapid and cost-effective validation and acceptance, rather than developing and validating in vitro test methods for individual cytokines, where a great many in vitro assays would be needed, likely all with specific data interpretation procedures and likely requiring licensing fees. Such complexity could be reduced without compromising data quality and human relevance.

The increasing understanding of the underlying mechanisms of tumor immunology, together with methods optimised for the identification of immunotoxicants, can offer a new strategy for the in vitro identification of chemical carcinogens. This understanding is fundamental for advancing regulatory frameworks pragmatically whilst improving the hazard and safety assessment of chemicals to protect public health.

## 9. Conclusions

The deregulation of the immune system plays a fundamental role in the mode of action of NGTxCs. Through interference with cytokine signalling, immune cell differentiation, immune surveillance, and inflammation pathways, NGTxCs contribute to immune dysfunction, a midway KE in the NGTxC IATA that alters the TME and that favours carcinogenesis.

In this analysis, we have summarised the key immune-related events underpinning NGTxCs’ effects, highlighting how specific biomarkers and functional immune readouts could be integrated into testing and assessment strategies for chemical hazard assessment purposes.

Understanding the complexity of immune responses and their modulation by chemical exposures is critical to developing robust, human-relevant models for the IATA.

The integration of immune dysfunction biomarkers into testing strategies has evolved greatly in recent years and is becoming essential to accurately capture the full continuum of key biological responses to NGTxC exposure from early adaptive responses to adverse outcomes.

Future efforts should focus on refining the characterisation of immune system perturbations and optimising cytokine marker combinatorial kits, both for hazard and diagnostic purposes. Whilst some cytokine-specific TGs have been developed, e.g., for IL-2 and IL8, the cost and time implications of developing and validating all the individual cytokine test methods may not be practical in the short term. Selective kit/cassette development, as conducted for diagnostic purposes, is more realistic, and there are many companies offering such services.

Improving the robustness and reproducibility of predictive models is an additional need with the application of omics and systems biology tools, as is establishing clear regulatory frameworks that recognise the importance of adverse immune modulation, immunotoxicity, immunosuppressionm and immune evasion. Altogether, such efforts will afford greater and more relevant carcinogenic risk assessment protection.

## Figures and Tables

**Figure 1 ijms-26-06310-f001:**
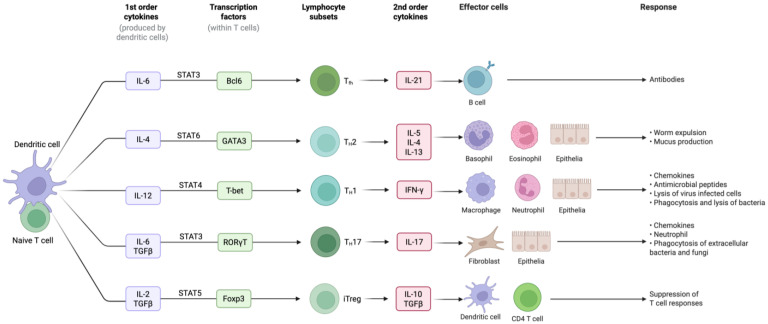
**Overview of the different roles of cytokines in shaping specific and coordinated immune responses.** Cytokines orchestrate immune responses by influencing multiple cellular processes, including the proliferation, differentiation, and effector functions of both innate and adaptive immune cells. Depending on their origin, timing, and receptor interactions, cytokines can promote pro-inflammatory or anti-inflammatory effects, regulate immune cell recruitment, polarise T helper cell responses (e.g., Th1, Th2, Th17, Treg), and maintain immune homeostasis. This schematic highlights the diversity of cytokine functions and their role in the dynamic regulation of immune pathways relevant to non-genotoxic carcinogenesis.

**Figure 2 ijms-26-06310-f002:**
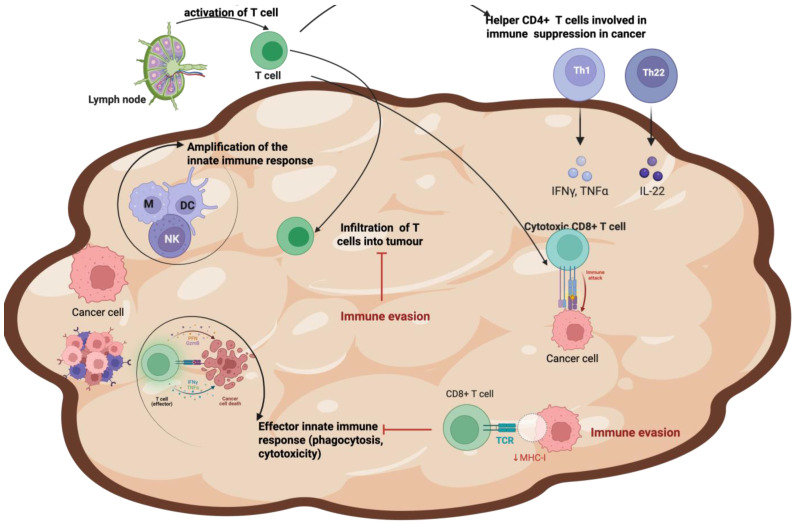
**Immune surveillance and evasion mechanisms within the tumor microenvironment.** This figure illustrates key steps in the immune defence response against tumor cells within the tumor. T cells primed in the lymph node infiltrate the tumor, where they interact with antigen-presenting cells such as macrophages (M), dendritic cells (DC), and natural killer (NK) cells, amplifying the innate immune response. Effector CD4^+^ and CD8^+^ T cells, including the Th1 and Th22 subsets, mediate anti-tumor functions through cytokines (e.g., IFN-γ, TNF-α, and IL-22) and direct cytotoxicity. However, tumor cells can evade immune attack by downregulating MHC-I, inducing T cell exhaustion, or modulating cytokine signalling.

**Figure 3 ijms-26-06310-f003:**
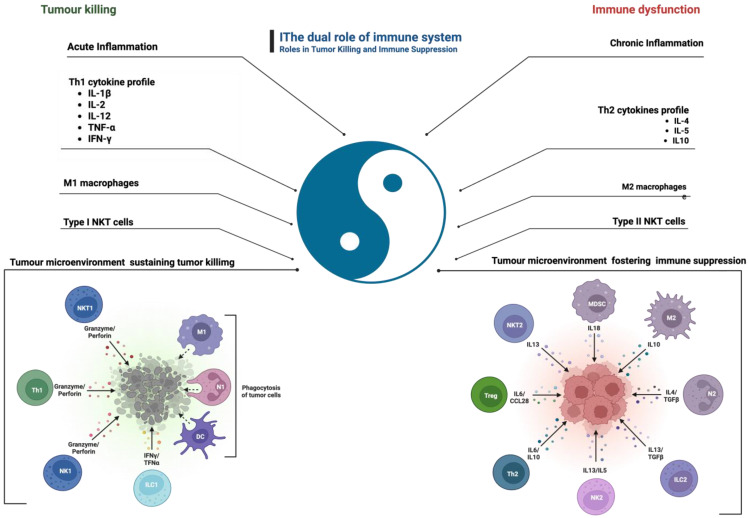
**The dual role of the immune system in cancer suppression and evasion.** The figure illustrates the complex duality of the immune system in cancer, highlighting its roles in both tumor suppression and tumor promotion. This complex interplay can be likened to a ‘yin-yang’ dynamic, where the same immune components can have opposing effects, depending on the context.

**Figure 4 ijms-26-06310-f004:**
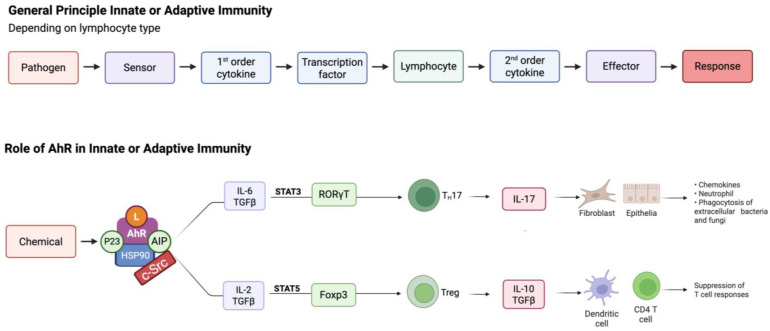
**Role of the AhR in modulating adaptive immune responses.** This figure illustrates how the activation of the AhR by chemical ligands can influence the differentiation of CD4-positive T cells (CD4+ T cells) into T helper 17 cells (Th17) or regulatory T cells (Treg). Through its interaction with different cytokine environments, AhR modulates the expression of key transcription factors: RAR-related orphan receptor gamma T (RORγT), promoting interleukin-17 (IL-17) production and pro-inflammatory responses, or Forkhead box P3 (Foxp3), promoting interleukin-10 (IL-10) and transforming growth factor-beta (TGF-β) production and immunosuppressive functions. These pathways impact tissue responses, including chemokine production, neutrophil recruitment, and the suppression of T cell activity, highlighting AhR’s critical role in immune homeostasis and cancer-related immune dysregulation.

**Figure 5 ijms-26-06310-f005:**
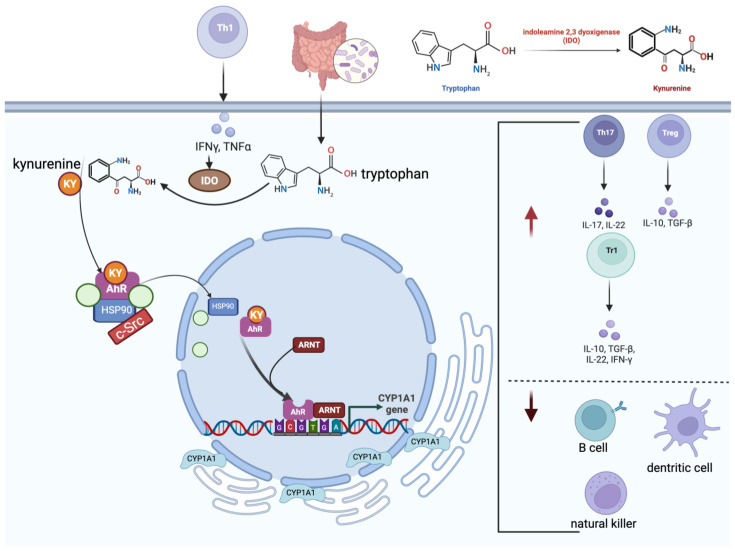
**Activation of AhR by kynurenine, leading to the differential modulation of T cell subsets.** This figure illustrates the activation of the AhR upon binding with kynurenine, a metabolite of the tryptophan pathway. Once activated, AhR promotes the differentiation and activation of various T cell subsets, particularly Th1, T regulatory (Treg), and Th17 cells. Simultaneously, the activation of AhR leads to a reduction in the activity and numbers of B cells, dendritic cells, and natural killer (NK) cells, indicating a shift towards a more immunosuppressive environment. This regulatory pathway is particularly relevant in immune tolerance, autoimmunity, and cancer, as well as in response to environmental stressors, including non-genotoxic carcinogens. On the right hand side of the Figure 5, the upward arrow indicates an increase, whilst the downward arrow indicated a reduction.

**Figure 6 ijms-26-06310-f006:**
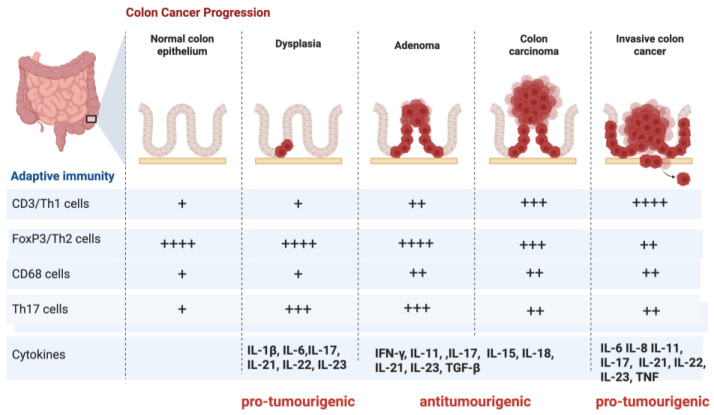
**Immune modulation/adaptation across the progressive stages of colon carcinogenesis.** This figure illustrates the dynamic evolution of adaptive immune responses during colon cancer development, from normal colon epithelium to invasive carcinoma. The ‘+’ symbols indicate the relative increase in the prevalence or functional influence of specific immune subsets or cytokine profiles at each stage. In the early phases (normal epithelium and dysplasia), the immune microenvironment shows a pro-tumorigenic profile that supports initial tumor growth. During the adenoma and carcinoma stages, the enhanced activation of anti-tumor immune mechanisms is observed, reflecting an attempt to suppress tumor progression. However, as cancer advances to invasive stages, immune dysfunction prevails, leading to a renewed dominance of pro-tumorigenic signals. This complex and gradual transition is characterised by overlapping immune features and a dynamic imbalance between tumor-promoting and tumor-suppressing mechanisms.

**Figure 7 ijms-26-06310-f007:**
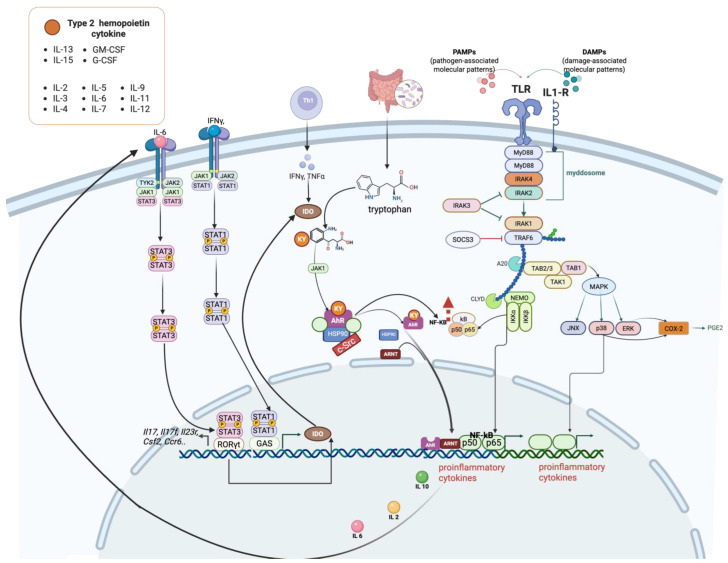
**AhR and JAK/STAT pathway crosstalk in immune regulation.** This figure illustrates the critical interaction between the AhR and the JAK/STAT signalling pathway in modulating immune responses, particularly through tryptophan metabolism. Upon activation by cytokines such as IFN-γ, the JAK/STAT pathway induces the expression of IDO, which catalyses the conversion of tryptophan into kynurenine. Kynurenine, in turn, activates AhR, forming a feedback loop where AhR modulates the JAK/STAT pathway and influences immune function. This loop plays a fundamental role in creating an immunosuppressive environment, particularly in the context of cancer, aiding in tumor immune evasion. Additionally, Toll-Like Receptor (TLR) signalling interacts with this system by activating the JAK/STAT pathway in response to microbial products, such as pathogen-associated molecular patterns (PAMPs) and damage-associated molecular patterns (DAMPs), expressed by host cells during cellular damage or cell death, further enhancing cytokine production and immune regulation. Upon activation, the TLR initiates a cascade that activates transcription factors such as NF-κB. TLR activation can modulate AhR activity and vice versa.

**Figure 8 ijms-26-06310-f008:**
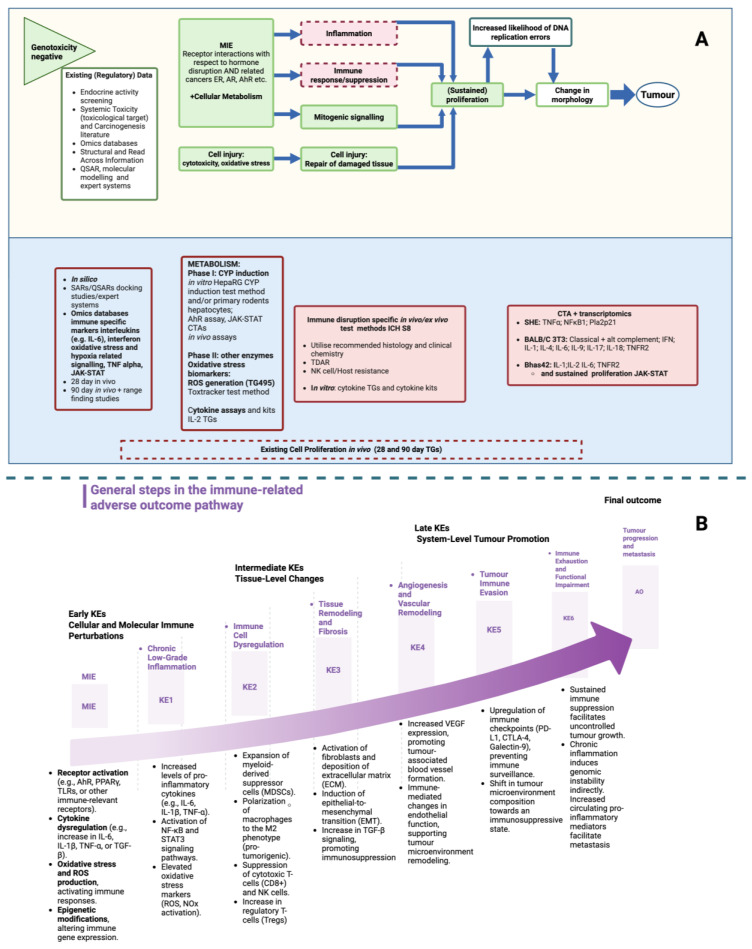
**Positioning immune disruption tools within the NGXC IATA.** Figure 8 illustrates how the immune-relevant biomarkers and test methods discussed herein can be incorporated into the NGTxC IATA. (**A**) Following a modular approach, in Module A, existing information and all relevant available regulatory data are reviewed to guide the testing priorities. This includes immune-related alerts that were identified from existing in vivo 28- and 90-day study data, rodent and EOGRTs (TG 443) from cohort 2, structural and read-across QSAR expert systems, screening data, etc. It also includes immune biomarker alerts identified from omics databases, including, e.g., interleukin cytokines for inflammatory responses, plus TNFα and JAK-STAT in relation to immune dysfunction, which have been proposed to support pre-screening [57]. For Modules B-C, a range of cytokines can be screened for and can also be detected variously in the different Cell Transformation Assay models [19] in Modules D-E [17]. (**B**) General steps in the immune-related adverse outcome pathway, aligned to the NGTxC IATA, outlining the key events (KEs) that link molecular initiating events (MIEs) to adverse outcomes in carcinogenesis.

**Table 1 ijms-26-06310-t001:** ICH S8 2006: Notes for guidance on immunotoxicity studies of human pharmaceuticals: Immune related parameters that should be measured in standard in vivo toxicology Test Guideline studies [54].

Parameter	Specific Component
Haematology	Total and absolute differential leukocyte counts
Clinical Chemistry	Unexplained alterations in globulin levels
Gross pathology	Lymphoid organs and tissues
Organ weights	Thymus, spleen (optional: lymph nodes)
Histology	Thymus, spleen, draining lymph node and at least one additional lymph node, bone marrow (particularly where unexplained alterations in peripheral blood or histopathological findings suggest cytologic evaluation of bone marrow) For oral administration: Peyers patch; for nasal/inhalation route: bronchus-associated lymphoid tissues (BALTs) and nasal-associated lymphoid tissues (NALTs)

**Table 2 ijms-26-06310-t002:** Selected chemicals used in the analysis of immunotoxicology relationships (extract modified from Luster et al. 1992).

Chemical	Immunotoxicity	Carcinogenicity	NK Cell Activity	T Cell Mitogens	MLR	CTL	Surface Marker
Allyl isovalerate	−	+	ND	−	ND	ND	ND
Azathioprine	+	+	−	−	−	+	+
Benzidine	+	+	+	+	+	ND	−
Benzo(a)pyrene	+	+	ND	+	−	ND	−
Benzo(e)pyrene	−	−	ND	-	ND	ND	ND
Cadmium chloride	+	+	−	+	ND	ND	ND
Cyclophosphamide	+	+	+	+	+	ND	+
4-chloro-o-phenylenediamine	−	+	−	−	−	ND	−
2,4-diaminotoluene	+	+	+	−	−	ND	+
Diethylstilbestrol	+	+	+	+	+	ND	+
Dimethylbenz[a]anthracene	+	+	+	+	+	+	ND
Dimethyl vinyl chloride	+	+	−	+	−	ND	ND
Diphenylhydantoin	+	+	+	−	ND	ND	−
Ethyl carbamate	+	+	ND	+	-	ND	ND
Ethylene dibromide	+	+	+	+	+	ND	ND
Formaldehyde	−	+	−	−	ND	ND	ND
Hexachlorobenzene-p-dioxin	+	+	−	−	+	ND	+
Methyl carbamate	−	+	ND	−	−	ND	ND
Nitrofurazone	−	+	−	−	−	−	−
n-nitro dimethylamine	+	+	+	+	+	ND	ND
Ochratoxin A	+	+	+	−	−	−	ND
Pentachlorophenol	+	+	+	+	−	ND	ND
o-phenyl phenol	+	−	ND	−	ND	ND	ND
Phorbol myristate acetate	+	+	ND	+	+	ND	ND
2,3,7,8-Tetrachlorodibenzo-p-dioxin (TCDD)	+	+	ND	ND	ND	ND	ND
Toluene	−	−	−	−	−	ND	−
4-Vinyl-1-cyclohexene diepoxide	+	+	−	−	−	ND	ND

Note: positive (+); negative (−); not done (ND); mixed leukocyte response (MLR); cytotoxic T lymphocytes (CTL). Chemicals for which rodent carcinogenicity classification was available were extracted from Luster et al. [47].

**Table 4 ijms-26-06310-t004:** **Classification of cytokines according to their role in the inflammation process supporting onco-immunity.** This table comprehensively captures all the essential immune biomarkers that are specifically relevant to NGTxC Integrated Approaches to Testing and Assessment (IATA).

Type of Cytokines (Classification)	Cytokines	Production Source	Primary Receptor	Key Function in Onco-Immunity	Reference
**Proinflammatory**	G-CSF	Fibroblasts, endothelium	G-CSFR	G-CSF promotes pro-tumorigenic immune phenotypes.	[135]
	IL-1α/IL-1β	Macrophages, B cells, DCs	IL-R1, IL-1R2 ^a^	IL-1α is activated downstream of oncogenic mutation, supporting tumor growth; IL-1β promotes angiogenesis.	[136]
	IL-6	Th cells, macrophages, fibroblasts	IL-6Rα IL-6ST (gp130)	IL-6 enhances cancer cell proliferation and survival by activating JAK/STAT; it creates an immunosuppressive TME by recruiting MDSCs and Tregs.	[137]
	IL-8	Macrophages	CXCR1, CXCR2	IL-8 is directly involved in EMT, increasing cell invasiveness and metastatic potential	[138]
	IL-11	Bone marrow stromal cells	IL-11Ra + IL-6ST	Stimulates angiogenesis of both primary tumors and metastatic sites. It is produced by CAFs in the tumor microenvironment.	[139]
	IL-17	Th17 cells	IL-17R (IL-17RA + IL17RC)	Il-17 acts as a key driver of tumor formation, growth, and metastasis by activating signal pathways (ERK, p38MAPK, NF-kB) in early tumorigenesis, orchestrating recruitment of Th17 cells, MDSCs and CAFs, stimulating endothelia cell migration and production of pro-angiogenic factors.	[140]
	IL-18	Monocytes, macrophages, DC	IL-1R5	IL-18 enhances cancer cell immune escape by suppressing CD70 expression and increasing metastatic ability through the upregulation of CD44 and vascular endothelial growth factor (VEGF). However, it can enhance IFN-γ production from T cells and NK cells, promoting anti-tumor immunity.	[141,142]
	IL-31	T Cells, monocytes, macrophages, DCs, mast cells, keratinocytes, fibroblasts	IL-31RA + oncostatin M receptor (OSMR)	IL-31 can promote tumor growth in some contexts, such as in follicular lymphoma, where it acts via autocrine/paracrine loops. Its anti-cancer effects are linked to anti-angiogenesis, the inhibition of tumor cell motility, and the enhancement of cytotoxic immune responses.	[143]
	IL-33	Lung epithelium and smooth muscle cells	IL-1R4	IL-33 is expressed by CAFs, promoting tumor growth, invasion, and metastasis. It can influence immune cells to create an immunosuppressive environment by Tregs and modulating T helper 17 (Th17) cells. IL-33 can enhance anti-tumor immunity by activating and recruiting immune effector cells such as CD8+ CTLs, NK cells, DCs, eosinophils, and group 2 innate lymphoid cells (ILC2s). This activation boosts the production of IFN-γ and TNF-α, promoting tumor cell killing and inhibiting tumor growth.	[144]
	IL-36A, B, G	36A: Spleen, lymph node, tonsils; 36B: bone marrow, tonsil, placenta; 36G: placenta	IL-1R6	IL-36 signalling in cancer cells can have pro-tumorigenic effects by increasing cancer cell proliferation, migration, and invasion, as seen in colorectal cancer and lung cancer cell lines. IL-36 receptor (IL-36R) activation in tumor cells induces the expression of pro-tumorigenic chemokines and immune checkpoint molecules such as PD-L1, which may facilitate tumor immune evasion. In the TME, IL-36 can promote anti-tumor immunity by enhancing the formation of tertiary lymphoid structures (TLSs), which facilitate dendritic cell-mediated tumor antigen presentation and T-cell priming, leading to increased infiltration and activation of CD8+ T cells, CD4+ memory T cells, and B cells.	[145]
	IFN-α	Macrophages, neutrophils, and some somatic cells	IFNAR1, IFNAR2	IFN-α exerts direct anti-tumor effects by inducing apoptosis and modulating tumor cell proliferation. It also orchestrates complex immune responses that enhance tumor elimination through the activation of innate and adaptive immunity. However, chronic or dysregulated IFN-α signalling can contribute to immune evasion and tumor progression, reflecting a nuanced role in carcinogenesis and cancer immunity.	[146]
	IFN-β	Fibroblasts	IFNAR1, IFNAR2	IFN-β acts as a tumor suppressor by inducing cell cycle arrest and senescence in cancer cells and enhances anti-tumor immunity by improving cancer cell recognition and killing by immune cells. Its presence is critical for controlling tumor growth and progression through immune surveillance mechanisms.	[146]
	IFN-γ	T Cells and NK cells	IFNG R1; IFNGR2	FN-γ serves as a pivotal cytokine in cancer immunology with a double-edged-sword effect. It promotes anti-tumor immunity by activating immune cells, enhancing tumor antigen presentation, and inducing tumor cell death. Conversely, tumors can co-opt IFN-γ signalling to evade immune destruction by upregulating immunosuppressive pathways and becoming resistant to IFN-γ’s effects.	[146]
	TNF-α	Macrophages	TNFR1	TNF-α can promote tumor growth and metastasis by sustaining inflammation and supporting tumor cell survival, yet it also participates in anti-tumor immune responses by activating immune effector cells and inducing apoptosis under certain conditions.	[147]
	TNF-β	T Cells	TNFR2	As with TNF-α, TNF-β may have a dual role in cancer, acting as both a tumor promoter and suppressor depending on the context.	[148]
	IL-10	T cells, B cells, macrophages	IL-10Rα and Il-10Rβ	IL-10 plays a complex, context-dependent role in cancer immune dysfunction, balancing both immunosuppressive and immunostimulatory effects. IL-10’s role varies by cancer type. It is protective in inflammation-driven cancers (e.g., colorectal) but tumor-promoting in others (e.g., advanced breast cancer). See text for further details.	[149]
	IL-12	T cells, macrophages, monocytes	IL-12Rβ1 and IL-12R β2	IL-12 induces the differentiation of T helper 1 (Th1) cells and stimulates the production of IFN-γ. It can reprogram or inhibit immunosuppressive cells in the tumor microenvironment such as TAMs and MDSCs, which are major contributors to tumor immune evasion. IL-37 suppresses both innate and adaptive immunity by inhibiting pro-inflammatory cytokine production and controlling inflammatory stimuli.	[150]
	IL-22	Activated T-cells and NK cells	IL-22R, IL-10RB	IL-22 is crucial for inflammation control, mucous production, and tissue regeneration, helping to repair damage and maintain tissue integrity. When IL-22 expression is dysregulated or chronically elevated, it can promote carcinogenesis by sustaining inflammation, enhancing tumor cell proliferation, and aiding immune evasion. IL-22 activates oncogenic signalling pathways such as STAT3, AKT, MAPK, and NF-κB, which promote tumor growth, survival, and metastasis.	[151]
	IL-37	B-cells, NK cells, and monocytes	IL-1R5; IL-18BP	IL-37 suppresses both innate and adaptive immunity by inhibiting pro-inflammatory cytokine production and controlling inflammatory stimuli. IL-37 modulates the tumor microenvironment by regulating local immunity and cell crosstalk. On the contrary, IL-37 can promote tumor immune evasion by inactivating cytotoxic CD8+ T cells through the IL-37/SIGIRR axis, reducing their proliferation and effector functions, which facilitates tumor escape from immune surveillance, as observed in colorectal cancer.	[152]
	IL-38	B cells and macrophages	IL-1R9; IL-1R6	IL-38’s function depends on the tumor type and microenvironment. IL-38 modulates the tumor immune microenvironment by suppressing pro-inflammatory cytokines and reducing recruitment and activation of cytotoxic T cells, thereby facilitating immune evasion in tumors such as lung, prostate, brain, and squamous cell carcinomas. It exerts anti-tumorigenic activity by inhibiting inflammatory signalling pathways, reducing tumor proliferation, and enhancing T cell-mediated immunity in colorectal cancer.	[152]
	TGF-β		TGF-βR1, 2, 3	In early cancer development, TGF-β inhibits tumor formation by inducing apoptosis in premalignant cells and blocking their proliferation and malignant transformation. As cancer progresses, some cancer cells evade the growth-inhibitory effects of TGF-β by decoupling its tumor-suppressive signals from processes like EMT in advanced cancers. See text for further details.	[153]
**Adaptive immunity**	GM-CSF	T cells, macrophages, fibroblasts	GM-CSFRα, GM-CSFRβ) (beta common chain)	GM-CSF modulates the immune system by driving the generation and activation of myeloid cells (neutrophils, monocytes, macrophages, and dendritic cells), which bridge innate and adaptive immunity. Its immunostimulatory effects contribute to anti-cancer functions by enhancing innate immune responses, promoting dendritic cell maturation, and activating T cells against tumor antigens. However, excessive GM-CSF can lead to immune cell exhaustion and dysfunction, impairing effective anti-tumor immunity and potentially promoting immune suppression within the tumor microenvironment.	[154]
	IL-2	Th1 cells, NKT, DCs and mast cells	Trimeric complex: IL-2RA, IL-2RB, and gamma chain (IL-2RG).	IL-2 deficiency contributes significantly to immune dysfunction in cancer by impairing T cell activation and proliferation. IL-2 promotes Treg expansion, which suppresses anti-tumor immunity. Tregs constitutively express IL-2 receptors and proliferate in response to IL-2, creating an immunosuppressive TME that aids tumor immune evasion.	[155]
	IL-3	T cells	IL-3Rα, IL-3Rβ (beta common chain)	Whilst IL-3 supports haematopoietic growth and can contribute to haematologic malignancies, its role in solid tumors is less well defined but increasingly appreciated for modulating immune cell recruitment and function within the tumor microenvironment.	[156,157]
	IL-4	T cells, mast cells, basophils and eosinophils	IL-4Rα	IL-4 has dual and context-dependent roles in cancer: It promotes carcinogenesis and tumor immune evasion by fostering an immunosuppressive microenvironment, enhancing tumor cell survival, and impairing cytotoxic immune cells. Conversely, IL-4 can also support anti-tumor immunity by revitalising exhausted T cells and synergising with existing immunotherapies. These paradoxical effects depend on factors such as IL-4 source, timing, dose, and the cellular and molecular tumor environment.	[155]
	IL-5	Th2 Cells and mast cells	IL-5Rα IL-5Rβ (common beta chain)	IL-5 contributes to carcinogenesis and cancer immune dysfunction mainly by shaping the tumor microenvironment through eosinophil and myeloid suppressor cell recruitment, promoting tumor progression and immune evasion.	
	IL-7	B and T cells, endothelial cells, bone marrow cells, epithelial cells	IL-7Rα and common gamma chain	IL-7 plays a dual role in cancer: **Tumor-suppressing role:** By enhancing immune cell function and promoting anti-tumor immunity, IL-7 can improve tumor eradication and is a potential immunotherapeutic agent. **Tumor-promoting role:** Through the activation of oncogenic signalling pathways, IL-7 can support tumor cell growth, survival, and metastasis, contributing to cancer progression.	[155]
	IL-9	T cells and mast cells	IL-9Rα (common gamma chain	IL-9 exhibits a dual role in cancer: **Tumor-promoting role:** In some cancers, especially lung cancer, IL-9 supports tumor survival and immune evasion by enhancing immunosuppressive cells and factors within the tumor microenvironment. **Tumor-suppressing role:** In many solid tumors, IL-9 produced by Th9 cells activates both innate (mast cells) and adaptive (CD8+ T cells) immunity, directly induces tumor cell apoptosis, and suppresses tumor growth.	[155]
	M-CSF	T cells and B cells	CSF-1R	M-CSF contributes to carcinogenesis by promoting tumor-supportive macrophage polarisation, enhancing tumor angiogenesis, and suppressing effective anti-tumor immune responses. Its modulation of both innate and adaptive immunity within the tumor microenvironment underlies its role in cancer immune dysfunction and highlights its potential as a target for therapeutic intervention.	[158]

Abbreviations: CAFs: cancer-associated fibroblasts; CTLs: cytotoxic T lymphocytes; DCs: dendritic cells; EMT = epithelial–mesenchymal transition; MDSCs: myeloid-derived suppressor cells; NK: natural killer cells; TAMs = tumor-associated macrophages; TME: tumor microenvironment; Treg = activating regulatory T cells. **Note:** Cytokine-receptor assignments for the IL-1 family members were updated according to the *International nomenclature guidelines for the IL-1 family of cytokines and receptors* [159]. Other cytokine-receptor assignments were verified using updated scientific databases [160,161,162,163] and recent immunology reviews [22,164] to ensure consistency with immunological knowledge to date. ^a^ Only the primary functional receptors for each cytokine are reported. The decoy receptor IL-1R2 has been included as it is reported among primary receptors in [159].

**Table 5 ijms-26-06310-t005:** Current regulatory inflammation/immune assay tools suitable for the NGTxC IATA, according to the modular approach [17].

Type of Assay	Reference	Mechanism	Regulatory Status * with Respect to NGTxC Applications
Module A Existing information Pre-Screening	[57]	Interferon and IL signalling, TNFα, JAK-STAT	*B: Needs a case study to establish it as an existing information/screening tool
In vitro			
Module B-C MIE IL-2 Luc Assay IL-8 Luc Assay MITA Assays	[55,58,129]	IL signalling	*A: Accepted OECD TG TG 444A TG 442E [58,165]: skin sensitisation. In use for cosmetics [58,129]
Module B Oxidative stress: ROS generation assay	[166]	ROS	In use for cosmetics *B-A
Module E CTAs	[19,63,167,168]	Persistence of Interferon and IL-signalling, TNFα, JAK-STAT, (CTA specific), IL-6 can be identified	OECD Guidance Documents *B-A: Needs validation work (laboratory transfer) for all 3 CTA models Reproduced within a laboratory for at least 3 chemicals
In vivo Modules A-E reviewed as part of existing information in Module A if data available, Modules C-D if testing needed.			
T-cell dependent antibody response (TDAR) immunosuppression	[60,169]	General Immunosuppression	*A: Accepted Chemicals/Pharma/Agchem, including similar markers in 28- and 90-day studies (OECD TG 407, 408) [51,52]
Natural Killer (NK) cell/Host resistance and others	[54]	Specific immune-cell response, ex vivo	*A: Accepted Pharma, including in 28- and 90-day studies (OECD TG 407, 408) [51,52]
**Critical assay/marker gaps**			
IL-6, IL-1, JAK-STAT, TGFβ	[19,57]	Cytokine biomarkers: Chemical selection needed to target specific markers.	Alerts from existing information, including 28- and 90-day studies (OECD TG 407, 408) and CTAs

* Regulatory readiness levels described in Jacobs et al. 2020 [13].

**Table 6 ijms-26-06310-t006:** Immunomodulation-driven AdverseOutcome Pathway (irAOP) for non-genotoxic carcinogens with associated biomarkers and test methods.

Stage	Description (Biological Mechanisms and Implication(s))
**MIE**	**Receptor activation** (e.g., AhR, PPARγ, TLRs, or other immune-relevant receptors). **Cytokine dysregulation** (e.g., increase in IL-6, IL-1β, TNF-α, or TGF-β). **Oxidative stress and ROS production** and activating immune responses. **Epigenetic modifications** and altering immune gene expression.
**Early KEs** **(Cellular and Molecular Immune Perturbations)**	**Chronic Low-Grade Inflammation** - Increased levels of pro-inflammatory cytokines (e.g., IL-6, IL-1β, TNF-α). - Activation of NF-κB and STAT3 signalling pathways. - Elevated oxidative stress markers (ROS, NOx activation). **Immune Cell Dysregulation** - Expansion of myeloid-derived suppressor cells (MDSCs). - Polarisation of macrophages to the M2 phenotype (pro-tumorigenic). - Suppression of cytotoxic T-cells (CD8+) and NK cells. - Increase in regulatory T-cells (Tregs) → immune evasion.
**Intermediate KEs (Tissue-Level Changes)**	**Tissue Remodelling and Fibrosis** - Activation of fibroblasts and deposition of extracellular matrix (ECM). - Induction of epithelial-to-mesenchymal transition (EMT). - Increase in TGF-β signalling, promoting immunosuppression. **Angiogenesis and Vascular Remodelling** - Increased VEGF expression, promoting tumor-associated blood vessel formation. - Immune-mediated changes in endothelial function, supporting tumor microenvironment remodelling.
**Late KEs** **(System-Level Tumor Promotion)**	**Tumor Immune Evasion** - Upregulation of immune checkpoints (PD-L1, CTLA-4, Galectin-9), preventing immune surveillance. - Shift in tumor microenvironment composition towards an immunosuppressive state. **Tumor Progression and Metastasis** - Sustained immune suppression facilitates uncontrolled tumor growth. - Chronic inflammation induces genomic instability indirectly. Increased circulating pro-inflammatory mediators facilitate metastasis.
	**Tumor promotion and progression** in **various tissues** (e.g., lung, liver, breast, prostate). **Increased risk of cancer development** in individuals exposed to NGTxCs through environmental, occupational, or pharmaceutical sources.
**Methods**	**Biomarkers**
**In vitro models**	**Human PBMC assays**: - Measuring cytokine profiles (IL-6, TNF-α). **Macrophage polarisation assays**: - Identifying M1/M2 shifts. **T-cell suppression assays**: - Evaluating immune evasion mechanisms. **3D co-culture systems**: - Modelling TME with immune cells and stromal components. **Transcriptomics and Epigenomics**: - Identifying immune gene expression changes.
**In vivo models**	**Humanised mouse models**: - Tracking immune suppression and inflammation. **Tumor progression models**: - Assessing the impact of chronic immune modulation. **Multi-omics approaches**: - Integrating proteomics, metabolomics, and lipidomics.
**Methods and biomarkers of human relevance (identified in human studies)**	**Systemic inflammation markers**: - IL-6, CRP, TNF-α. **Immune cell profiling**: - Tregs, MDSCs, exhausted T-cells (PD-1+ CD8+). **Soluble immune checkpoints**: - PD-L1, Galectin-9, CTLA-4. **Metabolomic signatures**: - Lactate, kynurenine, ROS metabolites.

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
