# Peer review of "Addressing Immune Response Dysfunction in an Integrated Approach for Testing and Assessment for Non-Genotoxic Carcinogens in Humans: A Targeted Analysis"

_ijms, 2025, doi:10.3390/ijms26136310_

Round 1

Reviewer 1 Report

Comments and Suggestions for Authors

The authors of this review elaborate how carcinogenic agents can induce cancerous genomic alterations through non-genotoxic carcinogenesis with a specific focus un immune dysfunction. For this purpose, the authors first explore the role of the immune system in cancer development and immunosurveillance. Key focusses of this review lie on the aryl-hydrocarbyl receptor (AhR) and the role of cytokines. Furthermore, the authors introduce key events and molecular initiating events in carcinogenesis and link these with cellular and molecular alterations. Lastly, important immune-oncology biomarkers, cytokines and detections tools/assays are discussed and placed within the regulatory landscape. 

Major comments:

  • Figure 2: This figure in my opinion adds nothing to the scope of the review as it is comparatively unspecific. Either major adjustments are necessary or this figure should be removed.
  • Figure 6: It is unclear to me, what information the “+” signs are supposed to convey. Is it the prevalence of this cell subset or the importance of this subset for the cytokine profile? Also, it is displayed in a way which makes the reader believe that cytokine profile in dysplasia and invasive colon cancer stages is very contrasting (pro-tumorigenic) to the profile in adenoma and carcinoma (anti-tumorigenic). However, the reality is much more complex although a disbalance towards pro- or anti-tumorigenic profiles might prevail. Here the authors should re-evaluate the information they want to convey with this figure.

Minor comments:

  • Citations and references should be in unified form. The authors switch between [XX] and (Author et al) format. E.g. Line 62. Additionally, multiple citations should be uniformly grouped within the same bracket “[XX; XX]” not listed separately “[XX]; [XX]”. The authors switch between both styles.
  • There are often symbols missing or superfluous: E.g. Line 41: “.” missing after “infectious agents [6]”; Line 263: Missing word “In the early 2000s, were made”; Line 374: Missing word “and reviewed in s [63].” and many more.
  • Line 78: Uniformity in abbreviations, either all capital or lower-case lettering. E.g. “Molecular Initiating Events (MIE) and key event (KE)”. Also some abbreviations are introduced twice, e.g. Line 89: “key event”, too late, e.g. “tumor microenvironment” line 444 or not at all, e.g. “Ahr” line 340.
  • Figure 2: Based on how the figure is structured, it appears that all mechanisms on the right side, e.g. interaction of CD8 T cells with cancer cells, happen outside of the tumor (as the tumor mass is on the left). This should be altered.
  • Also the labeling and lettering in many figures is too small to read, e.g. at the lower part within the tumor mass of Figure 2, as well as the whole of figures 3 and 7.
  • Line 196: Neither Fig. 2 nor Fig. 3 really highlight the three phases of the immunoediting hypothesis. The authors should remove the reference at this point or alter the figures accordingly.
  • Line 314: TNF-“α”, symbol not displayed correctly.
  • Line 832: The reference “Table 2“ should likely be “table 3”.
  • Table 3 contains a lot of important information but it is straining to read. I recommend to carefully reassess what information in the text is really relevant to the study and then display this as bullet points, not written out sentences. Patient number should be uniformly listed either in the “tissue” section (as for the first study) or in the “”study design” section. Also abbreviations should be used and explained in the text below the table, not within it.
  • Table 4 and 5A: No explanation given for why some text is in bold lettering.
Comments on the Quality of English Language

Moderate editing of English language recommended. General checking of missing or superfluous symbols recommended (e.g. missing “.”, missing brackets, etc.).

Author Response

Pease see the attachment

Reviewer 2 Report

Comments and Suggestions for Authors

Non-genotoxic carcinogens act as tumor promoters, endocrine modulators, receptor mediators, immunosuppressants, or inducers of tissue-specific toxicity and inflammatory responses. Deregulation of the immune system is one of the key actions associated with non-genotoxic carcinogens. In this review, the authors aim to summarize the key events underlying immune dysfunction mediated by non-genotoxic carcinogens. While the review effectively outlines the mechanisms of immune dysfunction and discusses an integrated approach to testing and assessment, several issues must be addressed to enhance the scientific validity of the claims. 

Major points:

(1) It is recommended to include legends for Figures 1, 2, 4, and 6 to enhance clarity.

(2) Figure 4

IL-2+TGF-beta ->  STAT3 ->  Foxp3 -> Treg

Please verify whether the information presented is accurate.

(3) Line739

The authors wrote that “In mice, IL-10 production is primarily associated with Th1 cells within the thymus, whereas in humans, both Th1 and Th2 cells contribute to IL-10 production [112].”.

T helper cells are known to differentiate in secondary lymphoid organs, but not in the primary lymphoid organ, the thymus. 

Please clarify whether the above sentence is accurate.

(4) It is advisable to add a “Conclusion”section at the end of the manuscript.

(5) Please review the contents of Table 4 carefully to ensure correctness.

IL-1 -> IL-1R1 and IL-1R3?

IL-18 -> IL-1R5 and IL-1R7?

IL-33 -> IL-1R3 and IL-1R4?

IL-36alpha -> IL-1R6 and IL-1R3?

IL-36beta -> IL-1R6 and IL-1R3?

IL-36gamma -> IL-1R6 and IL-1R3?

IL-37 -> IL-1R5 and IL-1R8?

IL-38 -> IL-1R6 or IL-1R9?

(Please see, Nat Immunol, 2024, 25, 581-582)

IL-8 -> IL-8RA (CXCR1) or IL-8RB (CXCR2)?

IL-17A to F differentially bind to IL-17RA to IL-17RE?

IFN-gamma -> IFNGR1 and IFNGR2?

TNF-beta -> TNFR1 or TNFR2 or HVEM?

IL-10 -> IL-10RA and IL-10RB?

IL-22 -> IL-22R and IL-10RB?

IL-12 -> IL-12Rbeta1 and IL-12Rbeta2?

TGF-beta -> TGFbetaR1 and TGFbetaR2?

IL-4 -> IL-4Ralpha + common gamma chain (CD132)?

        -> IL-4Ralpha + IL-13Ralpha1?

IL-11 -> e.g., hematopoiesis, etc.

IL-33 -> e.g., Th2 cytokine production, etc.

IL-7 -> T and B cell development, Treg development, memory T cell development, T cell homeostasis, etc.

IL-9 -> Th9 and Th2 development, mucus production, mast cell proliferation, etc.

IIL-2Rα -> IL-2Rα

IIL-3Rβ -> IL-3Rβ

Minor:

(6) Ref#1 is missing.

Please confirm whether this omission is intentional or an error.

(7) Line 59

Non-genotoxic carcinogens (NGTxC) are defined as chemicals that have the potential to induce cancer without interacting directly with either DNA or the cellular apparatus involved in the preservation of the integrity of the genome (Jacobs et al, 2020).

“Jacobs et al, 2020”? Ref#17?

(8) Line 94

Parallel synergistic efforts with respect to the key characteristics of immunotoxicity were identified to be alterations of cell signaling, immune cell proliferation, immune cell differentiation modulation, communication changes between immune cells and interference with immune cell trafficking (Germolec 2022).

“Germolec 2022”? Ref#13?

(9) Line 97

These all fall under the key hallmarks identified for NGTxC (Kravchenko et al 2015, Jacobs et al 2020).

“Kravchenko et al 2015, Jacobs et al 2020”? Ref#8 and 17?

(10) Line 1014

“Louekari and Jacobs 2024”? Ref#18?

(11) Line 133

The 2nd-order cytokines activate STAT3 and STAT1, and some also activate STAT2, which is associated with antiviral activity and STAT5.

Please clarify whether the description in the above sentence is accurate.

"STAT5"?

(12) Table 2

Regarding the abbreviation “TCDD”, please include its full spelling in Table 4.

(13) Line 314

TNF-?

(14) Line 512

Examples include interleukin-2 (IL-2), interleukin-4 (IL-4), interleukin-17 (IL-17), and interferon gamma (IFN-γ) IL32 is an example of a cytokine restricted to higher mammals, that is known to fine tune multiple pathways involved in metabolic processes or infection, and that promotes breast cancer cell invasion and metastasis via integrin β3-p38 MAPK signalling (see Table 2).

Please clarify whether the description in the above sentence is accurate.

(15) Line 729

“Table 5a” appears before “Table 3” and “Table 4”.  Please revise the order or clarify the rational behind this arrangement.

(16) Line 934

For modules B-C, a range of cytokines can be screened for, and can also be detected variously in the different Cell Transformation Assay models (Colacci et al 2023), in module D/E (Louekari and Jacobs 2024).

-> Module B-C and D/E?

Round 2

Reviewer 2 Report

Comments and Suggestions for Authors

#1

Comment from the reviewer

(2) Figure 4

IL-2+TGF-beta ->  STAT3 ->  Foxp3 -> Treg

Please verify whether the information presented is accurate.

Response from the authors

We thank the reviewer for pointing out this important correction. The figure has been updated to reflect that IL-2 activates STAT5, not STAT3, in the promotion of Foxp3 expression and Treg development. The caption has also been revised accordingly.

Comment from the reviewer

“IL-2” and TGF-beta are required for the induction of Foxp3, as shown in Figure 1.

#2

Comment from the reviewer

IL-1 -> IL-1R1 and IL-1R3?

IL-18 -> IL-1R5 and IL-1R7?

IL-33 -> IL-1R3 and IL-1R4?

IL-36alpha -> IL-1R6 and IL-1R3?

IL-36beta -> IL-1R6 and IL-1R3

IL-36gamma -> IL-1R6 and IL-1R3

IL-37 -> IL-1R5 and IL-1R8?

IL-38 -> IL-1R6 or IL-1R9?

(Please see, Nat Immunol, 2024, 25, 581-582)

IL-8 -> IL-8RA (CXCR1) or IL-8RB (CXCR2)?

IL-17A to F differentially bind to IL-17RA to IL-17RE?

IFN-gamma -> IFNGR1 and IFNGR2?

TNF-beta -> TNFR1 or TNFR2 or HVEM?

IL-10 -> IL-10RA and IL-10RB?

IL-22 -> IL-22R and IL-10RB?

IL-12 -> IL-12Rbeta1 and IL-12Rbeta2?

TGF-beta -> TGFbetaR1 and TGFbetaR2?

IL-4 -> IL-4Ralpha + common gamma chain (CD132)?

        -> IL-4Ralpha + IL-13Ralpha1?

IL-11 -> e.g., hematopoiesis, etc.

IL-33 -> e.g., Th2 cytokine production, etc.

IL-7 -> T and B cell development, Treg development, memory T cell development, T cell homeostasis, etc.

IL-9 -> Th9 and Th2 development, mucus production, mast cell proliferation, etc.

IIL-2Rα -> IL-2Rα

IIL-3Rβ -> IL-3Rβ

Response from the authors

Table 4 has been carefully reviewed and corrected in accordance with the most recent nomenclature for the IL-1 family (Nat Immunol, 2024, 25, 581–582) and additional updated scientific sources for other cytokine-receptor pairs Errors such as "IIL-2Rα" and "IIL-3Rβ" have been corrected 

Comment from the reviewer

Table 4 should be corrected consistently to ensure accuracy and uniformity.

For example, IL-1R1 is the primary receptor for IL-1alpha, although IL-1alpha binds to both IL-1R1 and IL-1R3. 

As noted in the reference (Nat Immunol, 2024, 25, 581–582), IL-1R2 functions as an inhibitory decoy receptor, as it is incapable of initiating downstream signaling. If the authors choose to list only the primary receptor, this is acceptable.

However, the primary receptor for IL-6 is IL-6Ralpha, which serves as the binding receptor. 

IL-6Ralpha cannot signal on its own and requires IL-6Rbeta (CD130, gp130) for downstream signaling. 

In the case of IL-10, IL-10Ralpha is the binding receptor, while IL-10Rbeta is also involved in signal transduction. 

Additionally, CDw210 should be corrected to CD210.

The receptors for IFNalpha are IFNAR1 and IFNAR2. 

CD118 is also a receptor for leukemia inhibitory factor (LIF).

TNF-alpha binds to both TNFR1 and TNFR2.

TNF-beta (lymphotoxin-alpha) binds to TNFR1, TNFR2, and HVEM (TNFRSF14).

doi: 10.1016/s1074-7613(00)80455-0

Typographical errors remain uncorrected in the table:

IL-2: “IIL-2Rα” should be corrected to “IL-2Rα”

IL-3: “IIL-3Rβ” should be corrected to “IL-3Rβ”

IL-7: Induces Th1 and Th17 response

IL-9: Promotes Th17 development

The above descriptions are inaccurate.

Please provide a more precise representation based on current immunological understanding.

The above descriptions serve as examples.

Please revise Table 4 to reflect consistent nomenclature and receptor classification throughout.

Author Response

Reviewer comment
“IL-2 and TGF-beta are required for the induction of Foxp3, as shown in Figure 1.”

Response from the authors
We thank the reviewer for highlighting this key aspect of Treg differentiation. We confirm that Figure 4 has been carefully revised and is now consistent with current scientific understanding. In the updated figure, IL-2 and TGF-β are both shown as required for Foxp3 expression. The aim of the figure is to provide a schematic overview of AhR-mediated immunomodulation, and we trust that the corrected version now accurately represents the pathway discussed in the manuscript.

Reviewer Comment: The table still contains errors and inconsistencies. The table should be more precise in receptor assignment, nomenclature, and function. Please clarify whether the goal is to show only the main signaling receptor or include also co-receptors and decoy receptors. Some cytokines (e.g., IL-6, IL-10, IL-7, IL-9) have oversimplified or inaccurate functions listed.

Response:

We thank the reviewer for this detailed feedback. As requested in the first-round review, Table 4 has already been aligned with the latest guidelines provided by Nature Immunology (Nat Immunol 2024, 25, 581–582). We also cross-checked cytokine–receptor assignments using curated resources such as the IUPHAR/BPS Guide to Pharmacology, UniProtKB, HGNC, and recent reviews.

In response to the current comment, we further clarified the nomenclature, and we have specified in the table legend that only primary receptors  are listed, to maintain consistency and avoid overcomplication. However, in the case of IL-1R2, we chose to retain it as a decoy receptor, explicitly labelling it as such in the table. This decision is based on its inclusion in the authoritative nomenclature table published by Nature Immunology (2024, 25, 581–582), to which the reviewer referred. We believe this ensures transparency and completeness while maintaining clarity on its lack of signalling function.

The descriptions of key functions  was presented in a simplified manner to serve the scope of the review and support integration in NGTxC-relevant IATA frameworks. Indeed, this table is not meant to cover cytokine biology exhaustively, but rather to support the IATA rationale by highlighting relevant functional associations.

For this purpose, a new column, ‘Key function in oncoimmunity,’ has been added to clarify the table’s relevance within the framework of non-genotoxic carcinogens  and immune dysfunction.

We hope that these adjustments and clarifications meet the reviewer’s expectations